# Morphological diversity within a core collection of subterranean clover (*Trifolium subterraneum* L.): Lessons in pasture adaptation from the wild

Abdi I. Abdi[1,¤], Phillip G. H. Nichols[1,2]*, Parwinder Kaur[1,2], Bradley J. Wintle[1,2], William Erskine[1,2]

**1** Centre for Plant Genetics and Breeding, School of Agriculture and Environment, The University of Western Australia, Crawley, Western Australia, Australia, **2** Institute of Agriculture, The University of Western Australia, Crawley, Western Australia, Australia

☯ These authors contributed equally to this work.
¤ Current address: Department of Plant Protection, College of Agricultural Engineering Sciences, The University of Duhok, Duhok, Kurdistan Region, Iraq
* phillip.nichols@uwa.edu.au

**Data Availability Statement:** All relevant data are within the paper and its Supporting Information files.

## Abstract

Subterranean clover (*Trifolium subterraneum* L.) is a diploid self-pollinated annual pasture legume native to the Mediterranean region and widely sown in southern Australia and other countries with Mediterranean-type climates. This study utilised a core collection of 97 lines, representing around 80% of the genetic diversity of the species, to examine morphological diversity within subterranean clover. A total of 23 quantitative agro-morphological and 13 semi-quantitative morphological marker traits were assayed on the core collection and 28 diverse Australian cultivars as spaced plants in a replicated common garden experiment. Relationships between these traits and 24 climatic and edaphic parameters at their sites of origin were also examined within the core collection. Significant diversity was present for all traits. The Australian cultivars had similar levels of diversity to the core collection for several traits. Among the agro-morphological traits, time to flowering, leaf size and petiole diameter in mid-winter, plant area in late winter, maximum stem length, content of the oestogenic iso-flavone biochanin A and total isoflavone content, were correlated with seven or more environmental variables. These can be considered highly adaptive, being the result of strong environmental selection pressure over time. For the first time in a clover species, morphological markers, including leaf mark, anthocyanin pigmentation and pubescence traits, have been associated with rainfall and soil parameters. This suggests they either have an adaptive role or the genes controlling them may be linked to other genes controlling adaptive traits. This study demonstrated the value of core collections to examine diversity within much larger global collections. It also identified adaptive traits from wild plants that can be utilised to develop more productive and persistent subterranean clover cultivars. The high heritability of these traits indicates that selection gains can be readily made.

**Funding:** AA conducted this research as part of his MSc studies at the University of Western Australia, funded by the Kurdistan Regional Government under the KRG-Scholarship program, Human Capacity Development (HCDP). Funding was also provided by Meat & Livestock Australia (MLA; www.mla.com.au), as part of the project 'Pre-breeding in annual legumes' (P.PBE.0037). PN and BW were employed by the Western Australian government in the Department of Primary Industries and Regional Development (DPIRD) at the time this research was conducted. The funders had no role in study design, data collection and analysis, decision to publish, or preparation of the manuscript.

**Competing interests:** The authors have declared that no competing interests exist.

## Introduction

Subterranean clover (*Trifolium subterraneum* L.) is a predominantly self-pollinated, diploid (2n = 16) annual pasture legume native to the Mediterranean basin, West Asia and the Atlantic coast of Western Europe [1, 2, 3, 4]. It has also been introduced to other regions of the world with Mediterranean-type climates, including southern Australia, South Africa, Chile, Argentina, the west coast and gulf regions of the United States of America, and to parts of New Zealand and Uruguay [5, 6, 7]. Of all the annual clovers, subterranean clover makes the greatest contribution globally to livestock feed production and soil improvement [5]. This is particularly the case in southern Australia, where it has been sown over 29 million ha and 53 cultivars have been released since 1900 [7, 8, 9].

The infra-specific taxonomy of *T. subterraneum* has been disputed. Katznelson and Morley [10] described three subspecies, *subterraneum*, *yanninicum* and *brachycalycinum*, based on plant morphology, genetic and cytogenetic data. Katznelson [2] later suggested the taxa *subterraneum*, *yanninicum* and *brachycalycinum* were separate species, while Zohary and Heller [3] classified *T. subterraneum* as a single species containing eight botanical varieties. However, more recent studies have supported the original classification into three subspecies, on the basis of isozyme patterns [11], karyotypes [12] and RAPD (random amplification of polymorphic DNA) markers [13]. In this paper, we have chosen to follow the original Katznelson and Morley [10] division of *T. subterraneum* into subspecies *subterraneum*, *yanninicum* and *brachycalycinum*. In their native habitat both ssp. *subterraneum* and ssp. *yanninicum* are found most commonly on moderately acidic soils, with ssp. *subterraneum* confined to well-drained soils and ssp. *yanninicum* to poorly drained soils, while ssp. *brachycalycinum* tends to grow on well-drained, neutral-alkaline, cracking or stony soils [2, 4, 14].

Diversity within subterranean clover is high for a range of phenological and growth traits [4, 7, 15, 16]. Flowering time is highly responsive to environment, with several studies demonstrating that early flowering is important for adaptation to short-season environments, with later flowering favoured in longer growing-season areas [15, 16, 17, 18, 19, 20]. Diversity has also been found for plant diameter, petiole length and thickness, leaf size, internode length, stem thickness, peduncle length and thickness and mean seed size [16, 18, 19, 21, 20]. Petiole length is an important factor determining competitive success within a pasture. Once canopy closure has occurred plants with long petioles intercept more of the incident light and shade out those with shorter petioles [22, 23, 24]. Large leaves are also important for light interception in competitive environments [15, 21]. Nichols et al. [16] showed that populations derived from a long growing-season area tended to have larger plants at maturity with larger leaves, longer, thicker petioles, thicker stems with longer internodes and longer, thinner peduncles than those derived from a short growing-season environment.

Fresh leaves of subterranean clover contain the isoflavones, formononetin, genistein and biochanin A, which are oestrogenically active and can cause infertility problems in ewes [25, 26, 27]. Of these, formononetin has the greatest effect. There is considerable diversity for the levels of each isoflavone, which vary independently of each other [4, 7, 16, 25].

Subterranean clover also has diversity for several other highly heritable morphological marker traits, which are used by seed certification authorities for cultivar identification [28]. These include seed colour, leaf marks, extent of leaflet indentation, intensity and pattern of leaf anthocyanin flushes and flecking, intensity and extent of stipule and calyx tube pigmentation and degree of pubescence of petioles, peduncles, stems and leaf upper surfaces [4, 7, 28].

A large global subterranean clover germplasm resource has been assembled from over 5,000 wild populations (hereafter referred to as accessions) collected in the Mediterranean basin and surrounding areas [4, 7]. The Department of Primary Industries and Regional

Development (DPIRD) in Perth, Western Australia have held the majority of these, with smaller collections held in genetic resource centres in Spain, Germany, Italy, New Zealand, Syria, the USA, Chile, Iran, Portugal and South Australia [4]. The DPIRD collection has recently been moved to the Australian Pastures Genebank, operated by the South Australian Research and Development Institute. These accessions have been sorted into more than 10,000 individual genotypes (hereafter referred to as lines) based on flowering times, morphological markers and isoflavone contents [4].

While the global subterranean clover collection is an excellent resource for plant breeding, its large size makes it unwieldy for efficient utilisation, particularly for investigations of trait diversity. In order to overcome this, a 'core' collection [29] of 97 distinct lines has been developed, which represents around 80% of the genetic diversity within the species [4, 7, 30]. This structured sample provides a research tool to investigate trait diversity within subterranean clover and the relationships between traits. As the location details of most wild collections have been recorded, associations can also be examined between trait expression among the core collection lines and attributes of their sites of origin, such as temperature and rainfall parameters, latitude, altitude, soil pH and texture. Such collection site information (referred to as 'passport data') can provide great insights into the relationships between environmental factors and expression of adaptive traits. It also contrasts with the situation for most diversity studies on grain crops, which rely on landraces and site data that is much less specific and has the confounding effects of human selection.

This study evaluated diversity for 23 agro-morphological and isoflavone traits related to adaptation and agronomic performance in the core collection of subterranean clover. It also examined, for the first time, diversity for 13 semi-quantitative morphological marker traits expected to be unrelated to adaptation. For comparison with the wild germplasm, the study also included 28 diverse Australian cultivars. Three main hypotheses are tested: (i) there is diversity for both agronomic and morphological marker traits in the core collection of subterranean clover; (ii) there is similar diversity to the core collection among the Australian cultivars; and (iii) specific traits are associated with climatic and edaphic parameters of their collection sites. It builds on the meta-analysis of Ghamkhar et al. [4] of diversity for flowering time and isoflavone content across the species, and spans an additional 31 traits. The study also provided an opportunity to validate core collections as a tool to examine diversity within global collections.

## Materials and methods

### Core collection lines and cultivars

The subterranean clover core collection was developed by K. Ghamkhar, R. Snowball and R. Appels [30]. From a total collection of over 10,000 lines obtained from genebanks around the globe, a subset of 760 clovers was selected based on diversity for both eco-geographical data from their sites of collection and agro-morphological data outlined by Ghamkhar et al. [4]. DNA was extracted from each member of this subset and 20 additional Australian cultivars. Forty eight SSR primers, spread across each of the eight subterranean clover chromosomes [31], were used to identify the most diverse germplasm among them. MSTRAT software [32] was used to optimise maximum diversity within the minimum number of lines. This resulted in an optimum number of 97 lines, covering around 80% of the genetic diversity within the global subterranean clover collection. Some of these lines were subsequently found to be mixtures of more than one genotype, based on leaf marks and other morphological traits, so single plant selections were made for each line to form the final core collection.

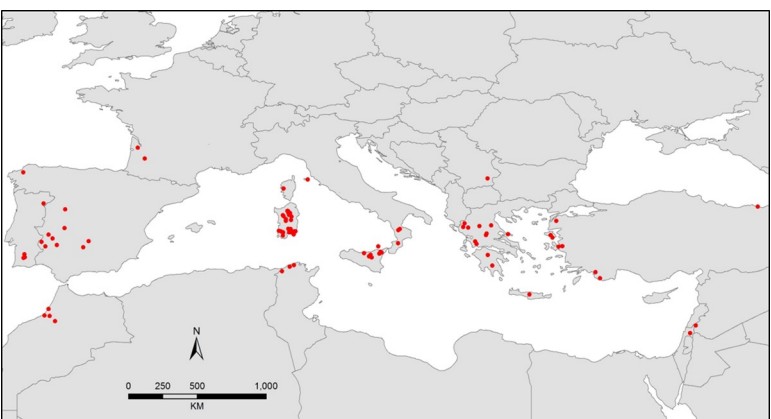

**Fig 1. Map of the Mediterranean region showing collection site locations for the 94 accessions of the subterranean clover core collection with known latitude and longitude.** Details of the individual lines collected at each site, along with other collection site information, are given in S1 Table.

The core collection consists of 95 lines derived from accessions collected from the Mediterranean region (designated L1 to L95) and two cross-bred cultivars, Coolamon (L96) [33] and Urana (L97) [34]. It comprises 66 lines of ssp. *subterraneum* and 31 lines of ssp. *brachycalycinum*, while there is no representation of ssp. *yanninicum*. These proportions approximate the ratio in the global subterranean clover collection of 66.3% ssp. *subterraneum*, 31.5% ssp. *brachycalycinum* and only 2.2% ssp. *yanninicum* [7].

The germplasm and associated passport data recorded by the plant collectors were obtained from the Australian Trifolium Genetic Resource Centre, formerly operated by DPIRD. Passport data were available for all 95 collected lines, apart from L79, which only had the country of origin. Fig 1 shows the geographical spread of the 94 core collection lines with known collection site, latitude and longitude. Details of the identity, latitude, longitude, altitude, soil pH and soil texture of each line is given in S1 Table. A summary of the passport data shows that ssp. *subterraneum* tended to be found at sites of higher altitude, lower pH and sandier texture than ssp. *brachycalycinum*, but both occurred at similar latitudes.

A set of 28 diverse Australian cultivars provided a comparison population to investigate diversity. It comprised five ssp. *yanninicum* cultivars (Gosse, Napier, Riverina, Trikkala and Yarloop) to overcome its lack of representation in the core collection, along with 19 ssp. *subterraneum* cultivars (Bindoon, Campeda, Daliak, Dalkeith, Denmark, Dinninup, Dwalganup, Geraldton, Goulburn, Izmir, Leura, Losa, Mt Barker, Narrikup, Nungarin, Rosabrook, Seaton Park, Woogenellup and York) and four ssp. *brachycalycinum* cultivars (Antas, Clare, Mintaro and Rosedale). Cultivar details are provided by Nichols et al. [7].

## Common garden experiment

The 97 core collection lines and 28 diverse cultivars were grown in a common garden experiment at the University of Western Australia Field Station, Shenton Park (31° 57' 3" S, 115° 47' 35" E). The trial had a row-column design with four replicates and five columns, each containing 100 plants, spaced 1 m apart (total of 500 plants). This spacing was sufficient to allow plants to grow without competition for light and moisture and prevented edge effects. An additional 50 plants, randomly selected from the core collection, were transplanted adjacent to the trial using the same spacing to provide material to calibrate shoot dry weight measurements.

Sowing date was May 15, 2014. Fifty seeds per line were placed on Whatman No 1 filter papers inside 10 cm diameter Petri dishes moistened with 3.5 mL of tap water at 15°C for 48

hours. This followed scarification to overcome seed coat dormancy using the technique of Nichols et al. [16]. Germinated seedlings then were planted into hydrated Jiffy-9 peat pots (Jiffy Products Ltd, Norway) and placed into sterilised seedling trays in the glasshouse. Two seedlings were sown per pot and randomly thinned to a single healthy seedling just prior to transplanting in the field. Group C *Rhizobium* (Nodulaid) was sprinkled onto peat pots 7 days after sowing (DAS). Peat pots were watered daily and soluble fertiliser (Thrive, Yates Australia) at a rate of 0.5 grams/L was applied weekly.

Seedlings were transplanted to the field on June 18 into a moist, weed-free seed bed, prepared by cultivation and prior application of 1 L/ha glyphosate herbicide (540 g a.i./L). Soil type was a deep sand with a pH (water) of 6.5 in the top 20 cm. Single superphosphate with potash (6.8% P, 12.4% K and 8.3% S) was drilled in just prior to transplanting at a rate of 180 kg/ha and was also applied by hand at the same rate on September 15. The site was hand-weeded throughout the experiment. Dimethoate insecticide (400 g/L a.i.) at a rate of 60 mL/ha was sprayed on August 26 to control blue-green aphids (*Acyrthosiphon kondoi*) and redlegged earth mites (*Halotydeus destructor*), while 150 mL/ha of alpha-cypermethrin (100 g/L a.i.) was sprayed on September 30 for native budworm (*Helicoverpa punctigera*) control. Irrigation was applied by overhead sprinklers as required until December 2. Mean monthly maximum and minimum temperatures for 2014 are shown in S1 Fig.

### Quantitative agro-morphological traits

Measurements were conducted on each plant, based on procedures described by Nichols et al. [16]. The first data were collected on August 11 (77 DAS), when all plants were still vegetative. This included maximum plant diameter, length and diameter (at the midpoint) of the longest petiole, and area of the largest leaf (three leaflets combined), which was measured using the leaf size plates and formulae of Williams et al. [35]. Petiole diameter was measured using a Mitotoyu® micrometer with 0.01 mm intervals.

On September 12 (110 DAS) plant height was measured and a 1–10 visual biomass rating was conducted, where 1 represents the lowest biomass of all plants and 10 the highest. The same biomass rating was conducted on the 50 calibration plants, which were then cut to ground level, oven-dried at 40°C for 48 hours and weighed. This led to derivation of the regression equation for dry weight: $y = 2.62x + 2.18$ ($R^2 = 0.73$), where y = plant dry weight and x = biomass rating. This equation was then used to convert the biomass ratings of all plants to dry weight.

Days to first flowering (DFF) was measured as the number of days from sowing in Petri dishes to the appearance of the first fully opened flower. Plants were checked every 3–4 days. Morphological characters related to flowering time were measured two weeks after first flowering. Leaf size, petiole length and petiole diameter were measured, using the same methodology as described above, on the leaf distended from the first flowering node on the longest stem. Peduncle length and diameter was also measured on this node. Internode length and stem diameter were measured between the first and second flowering nodes on the longest stem. Petiole, peduncle and stem diameters were measured using a Mitotoyu® micrometer, as described above.

Length of the longest stem was measured at senescence (determined when no green plant material remained). Following senescence plants were cut at ground level, burrs were removed by hand, and plant tops were oven dried at 40°C and weighed. Soil was dug to a depth of 2.5 cm to recover remaining buried burrs and sieved over a 1.6 mm mesh sieve to remove soil. Buried and unburied burrs were bulked, and seeds were extracted by a mechanical thresher, followed by hand removal of debris. Mean seed weight was calculated from a random sample of 100 seeds from each plant.

## Semi-quantitative morphological marker ratings

Morphological marker traits examined included leaf marks and their position, extent of leaflet indentation at the distal margin, intensity of leaf anthocyanin flecking and flush patterns, extent of stipule and calyx pigmentation, and pubescence of petioles, leaf upper surfaces, stems and peduncles. Rating scales are provided in S2 Table, using descriptors in Nichols et al. [28] and Ghamkhar et al. [4]. Ratings were conducted on plants in the first two replicates and checked against plants in the third replicate. Ratings for leaf and stipule characters were conducted on September 5, while those for flower and stem characters were conducted two weeks after the commencement of flowering.

Leaf marks are classified on the basis of: (i) width of 'crescents', which are central triangular markings (usually pale green) that extend from the leaflet centre towards the margins and range in size from a central dot ($C_1$) to all the way from margin to margin ($C_4$); (ii) breadth of 'arms', which are typically white or pale green and extend from the leaflet margins about halfway towards their centres and vary in breadth from narrow ($A_1$) to very broad ($A_4$); and (iii) breadth of 'bands', which are an alternative marking to crescents and arms, consisting of a pale green stripe extending from margin to margin and varying in breadth from narrow ($B_1$) to broad ($B_3$) [28]. Leaf marks can consist of crescents alone, arms alone, crescents with arms, bands alone or no marking at all. Leaf marks are positioned in most genotypes in the centre of leaflets but are positioned distal or proximal to this in some genotypes. Leaflet indentation of the distal margin also varies between genotypes, with ratings ranging from 0 (no indentation) to 4 (strong).

Anthocyanin flush patterns and leaf flecks are additional leaf markings, which vary in location, colour (usually dark purple or brown) and extent among genotypes. They are most prominent in winter and early spring and disappear as spring temperatures increase [28]. The extent of stipule anthocyanin pigmentation (usually purple) under closed canopies is also strongly genotype dependent [28]. The rating scale for anthocyanin intensity for these traits varies from 0 (absent) to 6 (strong). For calyx pigmentation, the amount and colour (shades of pink, red or purple) on the distal ends of calyx tubes is consistent among the flowers of individual genotypes for the duration of flowering and its extent varies from none (0 rating) to its entire length (8 rating) [28].

Pubescence of petioles, leaf upper surfaces, stems and peduncles varies independently, ranging from glabrous (0 rating) to very strong (8 rating), and is also strongly genotype dependent [28].

## Oestrogenic isoflavone contents

Levels of the oestrogenic isoflavones, formononetin, genistein and biochanin-A were measured in a separate experiment sown at the same site in 2011. The trial consisted of four replicates of the 97 core collection lines and the same 28 diverse cultivars grown as randomised spaced plants in a row-column design. Seeds were sown directly into Jiffy pots in the glasshouse on May 16 and transplanted to the field on July 6. Plant and row spacing and trial management were the same as for the 2014 experiment.

Isoflavone levels were measured using the thin-layer chromatography technique of Francis and Millington [25]. This method was used due to its high throughput and low cost, while recognising its semi-quantitative nature imposes limitations in precision. This method has also been previously used in similar genetic diversity studies by Nichols et al. [16] and Ghamkhar et al. [31]. Leaf samples of six discs (6 mm diameter) were taken from healthy, young leaves from Aug 8–31. Isoflavones were extracted in ethanol immediately and subjected to thin-layer

chromatography. Duplicate samples for dry weight determination were oven-dried at 60˚C for 48 hours. Isoflavone results are expressed as a percentage of dry matter (d.m.).

### Trait associations with collection site climate and soil data

Data for 19 quantitative temperature and precipitation (BIOCLIM) variables, derived from Hijmans et al. [36] and WorldClim [37], were obtained for 93 core collection lines. These 19 BIOCLIM variables are listed in S3 Table and described in more detail by Ghamkhar et al. [4]. BIOCLIM values for each of the 93 lines are given in S5 Table. BIOCLIM data was not available for L12 and L79, while L96 and L97 are cross-bred cultivars. Associations between traits and each BIOCLIM variable were evaluated by correlation for the 93 core collection lines with available data. The BIOCLIM rainfall variables were used in these analyses, in preference to estimates by the original collectors, as the dataset is standardised and more complete and several rainfall parameters can be examined. However, latitude, longitude, altitude, soil pH (H₂O), and soil texture data acquired from collectors' passport information [S1 Table] were included in analyses. Latitude and longitude data were available for all lines, apart from L79, L96 and L97, while availability of other passport data varied between lines, as shown in S1 Table. Soil texture was classified into five categories as 1 = sand, 2 = sandy loam, 3 = loam, 4 = clay loam and 5 = clay.

### Statistical analyses

Analyses of variance (ANOVAs) were used to compare the means for each of the 125 genotypes using GenStat 18th edition (VSN International Ltd, UK). ANOVAs were also conducted on genotype mean data to compare means of the core collection with that of the cultivars and to compare subspecies differences among the 125 genotypes. Residuals were analysed for assumptions of normality and homogeneity and square root or log transformations were conducted where appropriate when these assumptions were not fulfilled (see S4 Table).

Pearson correlation coefficients were calculated between agronomic traits and between morphological marker traits for each of the 125 genotypes. For the 94 core collection lines with known collection site locations Pearson correlation coefficients were also calculated between both agronomic and morphological marker traits and site of origin parameters. Excluded from analyses were the leaf mark position trait, due to the low number of states, and the arm colour and seed colour traits, due to non-linearity.

Broad-sense heritability (H²) was estimated for each quantitative trait as the percentage of the phenotypic variance attributable to genotypic variance, in accordance with Falconer et al. [38].

## Results

Table 1 provides a summary of genotype means, standard errors, minima, maxima and broad-sense heritability (H²) of all 23 agro-morphological and isoflavone traits for the 97 core collection lines and 28 diverse cultivars. Mean data for each of the 125 genotypes are shown in S6–S9 Tables.

### Diversity for flowering time

Flowering time differences were highly significant (P ≤ 0.001) among the 125 genotypes (Table 1 and S7 Table). The range in flowering time was greater among the core collection (86–176 DAS) than the cultivars (83–141 DAS) (Fig 2), but the means between the two groups

**Table 1. Summary of genotype means, standard errors (SE), minima, maxima and broad-sense heritabilities (H²) of quantitative agro-morphological characters measured for 97 core collection lines and 28 diverse Australian cultivars.** Differences were highly significant ($P \leq 0.001$) for all traits among the 125 genotypes.

| Trait | Core collection lines | | | | | Cultivars | | | | |
|---|---|---|---|---|---|---|---|---|---|---|
| | Mean | SE | Min | Max | H² (%) | Mean | SE | Min | Max | H² (%) |
| *At 77 DAS* | | | | | | | | | | |
| **Maximum plant diameter (cm)** | 24.7 | 0.45 | 14.0 | 36.6 | 66.2 | 24.1 | 0.86 | 16.9 | 33.0 | 67.2 |
| **Leaf size (cm²)** | 5.2 | 0.22 | 2.0 | 13.6 | 79.7 | 5.2 | 0.43 | 1.9 | 11.7 | 80.5 |
| **Petiole length (cm)** | 8.5 | 0.19 | 4.0 | 12.9 | 57.2 | 8.4 | 0.36 | 5.4 | 12.0 | 56.7 |
| **Petiole diameter (mm)** | 1.18 | 0.02 | 0.85 | 2.00 | 66.7 | 1.19 | 0.04 | 0.83 | 1.90 | 70.4 |
| *At 110 DAS* | | | | | | | | | | |
| **Plant height (cm)** | 6.6 | 0.22 | 2.0 | 11.4 | 54.5 | 6.8 | 0.50 | 3.6 | 15.6 | 64.4 |
| **Leaf size (cm²)** | 15.5 | 0.36 | 4.8 | 21.9 | 68.1 | 15.2 | 0.68 | 8.7 | 20.6 | 68.2 |
| **Biomass (g)** | 17.7 | 0.33 | 8.7 | 28.4 | 65.0 | 17.5 | 0.55 | 11.4 | 23.1 | 58.4 |
| **Plant area (cm²)** | 1250 | 71.7 | 172 | 4293 | 70.2 | 1268 | 121.6 | 390 | 2713 | 65.1 |
| *At first flowering* | | | | | | | | | | |
| **Flowering time (days)** | 119.8 | 2.04 | 86.0 | 176.0 | 98.4 | 112.6 | 3.23 | 83.0 | 141.0 | 97.8 |
| **Leaf size (cm²)** | 9.0 | 0.39 | 2.2 | 18.6 | 73.3 | 9.1 | 0.69 | 2.7 | 17.5 | 71.0 |
| **Petiole length (cm)** | 8.4 | 0.33 | 2.2 | 16.2 | 68.5 | 9.7 | 0.57 | 3.8 | 16.4 | 64.0 |
| **Petiole diameter (mm)** | 1.45 | 0.03 | 0.81 | 2.08 | 65.7 | 1.56 | 0.05 | 0.92 | 2.14 | 67.6 |
| **Internode length (cm)** | 5.1 | 0.17 | 1.7 | 9.5 | 72.4 | 4.8 | 0.28 | 2.5 | 9.6 | 65.9 |
| **Stem diameter (mm)** | 2.50 | 0.03 | 1.63 | 3.37 | 67.7 | 2.61 | 0.05 | 1.97 | 3.23 | 53.5 |
| **Peduncle length (mm)** | 56.1 | 2.60 | 24.3 | 132.0 | 69.1 | 51.1 | 4.76 | 29.0 | 125.8 | 72.8 |
| **Peduncle diameter (mm)** | 1.01 | 0.02 | 0.61 | 1.47 | 50.7 | 1.04 | 0.03 | 0.73 | 1.34 | 53.2 |
| *At senescence* | | | | | | | | | | |
| **Maximum stem length (cm)** | 63.9 | 2.10 | 10.3 | 146.0 | 88.3 | 66.5 | 3.70 | 29.7 | 104.0 | 87.8 |
| **Biomass (g)** | 156.6 | 8.5 | 17.0 | 402.9 | 92.1 | 152.6 | 11.3 | 57.7 | 247.9 | 84.9 |
| **Mean seed weight (mg)** | 8.3 | 0.26 | 5.2 | 20.8 | 98.5 | 8.6 | 0.33 | 4.4 | 11.3 | 96.4 |
| *Isoflavones (% of dry matter)* | | | | | | | | | | |
| **Formononetin** | 0.25 | 0.04 | 0.00 | 2.08 | 89.0 | 0.27 | 0.09 | 0.00 | 2.00 | 94.1 |
| **Genistein** | 1.17 | 0.06 | 0.10 | 2.58 | 76.4 | 1.18 | 0.11 | 0.25 | 2.28 | 70.2 |
| **Biochanin A** | 0.81 | 0.07 | 0.00 | 2.73 | 85.5 | 0.69 | 0.09 | 0.04 | 1.89 | 82.4 |
| **Total isoflavones** | 2.23 | 0.07 | 0.46 | 4.19 | 65.7 | 2.13 | 0.15 | 0.43 | 3.78 | 79.6 |

did not differ. Broad-sense heritability for flowering time was very high (98%) among both core collection lines and the cultivars (Table 1).

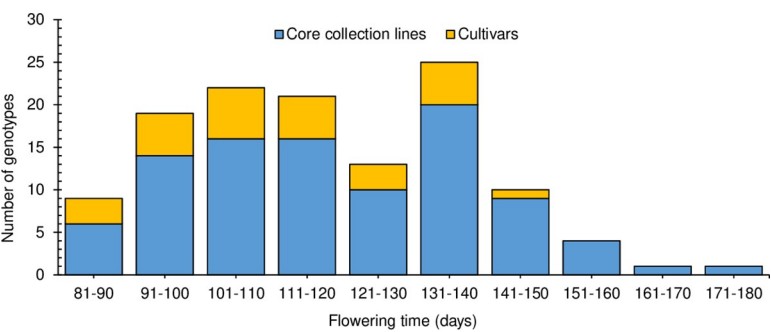

**Fig 2. Frequency distribution of mean flowering times (days from sowing to appearance of the first open flower) at Shenton Park, Western Australia of 97 subterranean clover core collection lines and 28 diverse Australian cultivars.**

**Table 2. Mean values of quantitative agro-morphological traits based on subspecies for the 97 subterranean clover core collection lines and 28 diverse Australian cultivars.** Subspecies with a common letter for each trait are not significantly different ($P < 0.05$). The significance level across all 125 genotypes is also shown for each trait.

| Trait | Subspecies mean values | | | Significance |
|---|---|---|---|---|
| | *subterraneum* | *brachycalycinum* | *yanninicum* | level |
| | n = 85 | n = 35 | n = 5 | |
| *At 77 DAS* | | | | |
| Maximum plant diameter (cm) | 22.8a | 29.0b | 25.1a | *** |
| Leaf size (cm²) | 4.4a | 7.2b | 4.9a | *** |
| Petiole diameter (mm) | 1.10a | 1.39b | 1.23ab | *** |
| Petiole length (cm) | 7.6a | 10.3b | 9.3b | *** |
| *At 110 DAS* | | | | |
| Plant height (cm) | 5.9a | 8.1b | 8.5b | *** |
| Leaf size (cm²) | 14.5a | 17.7b | 15.9ab | *** |
| Plant area (cm²) | 1021a | 1854b | 1014a | *** |
| Biomass (g) | 16.6a | 20.4b | 16.7a | *** |
| *At first flowering* | | | | |
| Flowering time (days) | 113.8a | 129.3b | 114.4ab | *** |
| Leaf size (cm²) | 8.1a | 11.1b | 10.3ab | *** |
| Petiole length (cm) | 8.9ab | 7.8a | 10.8b | ** |
| Petiole diameter (mm) | 1.41a | 1.57b | 1.85c | ** |
| Internode length (cm) | 4.7a | 6.1b | 4.0a | *** |
| Stem diameter (mm) | 2.55b | 2.41a | 2.79b | * |
| Peduncle length (mm) | 43.6a | 85.9b | 42.4a | *** |
| Peduncle diameter (mm) | 1.05b | 0.92a | 1.10b | *** |
| *At senescence* | | | | |
| Maximum stem length (cm) | 61.9a | 70.7b | 63.9ab | ** |
| Total biomass (g) | 136a | 202b | 154ab | *** |
| Mean seed weight (mg) | 8.5ab | 7.8a | 10.1b | ** |
| *Isoflavones (% of dry matter)* | | | | |
| Formononetin | 0.29b | 0.13a | 0.50b | ** |
| Genistein | 0.99a | 1.55b | 1.59b | *** |
| Biochanin A | 1.03b | 0.21a | 0.55b | *** |
| Total isoflavones | 2.31b | 1.90a | 2.64b | ** |

$^{*}P \leq 0.05$

$^{**}P \leq 0.01$

$^{***}P \leq 0.001$

Among the 97 core collection lines, mean flowering time of ssp. *brachycalycinum* lines was significantly later than ssp. *subterraneum* lines (Table 2). However, ssp. *subterraneum* had a greater range in flowering time (86–176 days) than ssp. *brachycalycinum* (100–156 days) (see S7 Table).

## Diversity for other quantitative traits

Considerable diversity was observed for all other agro-morphological and isoflavone traits (Table 1, S6–S8 Tables). Significant differences were found for all traits among the 125 genotypes (Table 2). Further ANOVAs showed no significant differences between the core collection and cultivar means for any traits, apart from petiole length and diameter at flowering, stem diameter and maximum stem length (discussed below).

Among measurements taken at 77 DAS, mean plant diameter of the core collection lines ranged 2.6-fold, size of the largest leaf ranged 6.8-fold, petiole length ranged 3.2-fold and petiole diameter ranged 2.4-fold (Table 1 and S6 Table). Mean cultivar values were within these ranges, apart from Daliak, which had the smallest leaf size (1.9 cm$^2$) and petiole diameter (0.83 mm) of all genotypes.

Of the measurements taken at 110 DAS, mean plant height among the core collection lines ranged 5.7-fold, size of the largest leaf ranged 4.6-fold, biomass per plant ranged 3.3-fold and plant area ranged 25-fold (Table 1 and S6 Table). Mean cultivar values were within these ranges, apart from the ssp. *yanninicum* Gosse, which had the greatest plant height (15.6 cm) of all genotypes.

At first flowering, mean leaf size among the core collection lines ranged 8.5-fold, petiole length ranged 7.4-fold, petiole diameter ranged 2.6-fold, peduncle length ranged 5.5-fold and peduncle diameter ranged 2.4-fold, while internode length ranged 5.6-fold and stem diameter ranged 2.1-fold (Table 1 and S7 Table). Cultivar values were within these ranges, apart from the ssp. *yanninicum* Napier, which had the thickest petioles (2.14 mm) and the ssp. *brachycalycinum* Clare, which had the longest internodes (9.6 cm) of all genotypes. Mean petiole length and petiole diameter of the cultivars were significantly greater than the core collection lines ($P \leq 0.001$) and stems were significantly thicker ($P \leq 0.01$).

At senescence, mean plant biomass among the core collection lines varied enormously, ranging from 17–403 g, while length of the longest stem varied from 10–146 cm and mean seed weight varied from 5.2–20.8 mg (Table 1 and S7 Table). Biomass and stem lengths of the cultivars were within these ranges, but Goulburn had lighter seeds (4.4 mg) than all other genotypes. However, stems of the cultivars were significantly longer on average ($P \leq 0.05$) than core collection lines.

Isoflavone contents among the core collection ranged from 0.0–2.1% of d.m. for formononetin, 0.1–2.6% of d.m. for genistein and, 0.0–2.7% of d.m. for biochanin A, while total isoflavone contents ranged from 0.5–4.2% of d.m. (Table 1 and S8 Table). Cultivar values were within each of these ranges, apart from Dalkeith, which had the lowest total isoflavone content of all genotypes (0.4% of d.m.). Mean values for the cultivars and core collection lines did not differ for any of the isoflavone traits. A closer examination of formononetin content showed that 28% of core collection lines had >0.20% of d.m., a level considered non-oestrogenic [7, 26, 27], compared to 21% of cultivars.

All traits were highly heritable, with broad-sense heritability values (H$^2$) > 50%, among both core collection lines and the cultivars (Table 1). The highest H$^2$ values, apart from flowering time, were for mean seed weight, maximum stem length, mature plant biomass and the three oestrogenic isoflavones.

## Subspecies comparisons

Significant subspecies differences were found for all agro-morphological and isoflavone traits among the 125 genotypes (Table 2). At 77 DAS ssp. *brachycalycinum* had larger plants with larger leaves than the other two subspecies, while ssp. *subterraneum* had shorter petioles than the other subspecies and thinner petioles than ssp. *brachycalycinum*. At 110 DAS, ssp. *subterraneum* had shorter plants than the other subspecies and smaller leaves than ssp. *brachycalycinum*, while ssp. *brachycalycinum* had bigger plants with higher biomass than the other two subspecies (Table 2).

Mean flowering time of ssp. *subterraneum* was similar to ssp. *yanninicum*, while ssp. *brachycalycinum* was later flowering. At first flowering ssp. *brachycalycinum* had larger leaves than ssp. *subterraneum* and had thinner stems with longer internodes and longer, thinner

peduncles than both subspecies, while ssp. *yanninicum* had longer petioles than ssp. *brachycalycinum* (Table 2). Petiole dimeter of ssp. *yanninicum* was greater than ssp. *brachycalycinum*, which in turn was greater than ssp. *subterraneum*. Fully mature plants of ssp. *brachycalycinum* had longer stems and greater biomass than ssp. *subterraneum*, while mean seed weight of ssp. *yanninicum* was significantly higher than ssp. *brachycalycinum* (Table 2).

Among the oestrogenic isoflavones, ssp. *brachycalycinum* had lower formononetin and biochanin A and lower total isoflavone contents on average than the other two subspecies, while ssp. *subterraneum* had lower genistein content (Table 2).

## Diversity for semi-quantitative morphological markers

Wide diversity was observed in each of the 13 morphological marker traits scored among both the core collection lines and the cultivars (S9 Table). In most cases, both the core collection lines and cultivars spanned the rating scales for the traits measured.

A frequency distribution of leaf mark types among core collection lines and cultivars is shown in S2 Fig. The most common leaf mark among both core collection lines (73.2%) and cultivars (60.7%) was a crescent with arms. This was followed by a crescent only, with 15.5% of core collection lines and 28.6% of cultivars. Among genotypes with crescent leaf marks, narrow crescents (3–4 rating) were most common, with 38% and 39% for cultivars and core collection lines, respectively (S9 Table). Among genotypes with arm leaf marks, moderately broad arms (3–4 rating) were the most common among core collection lines (42.3%), while narrow arms (1–2 rating) were most common among the cultivars (32.1%) (S9 Table). Among genotypes with leaf marks, 91.4% of core collection lines and 92.9% of cultivars were located centrally on the leaflet (S9 Table). Core collection lines spanned all categories for leaflet indentation (S3 Fig), with the highest frequencies being for moderate indentation (36.1%) and no indentation (35.1%). The highest proportion (57.1%) of cultivars had no indentation, while only one cultivar (Woogenellup) was strongly indented.

Frequency distributions of ratings for pubescence of upper leaf surfaces, petioles, stems and peduncles are shown in S4 Fig. Wide diversity was found among both core collection lines and cultivars. For leaf upper surfaces, the highest proportion of genotypes were glabrous among both core collection lines and cultivars. For petioles, 57.7% of core collection lines and 50.0% of cultivars had weak to moderate pubescence (2–4 rating), while 32.1% of cultivars had glabrous petioles. For stems, the highest proportion (41.2%) of core collection lines had moderately strong or strong stem pubescence (5–6 rating), while the highest proportion of cultivars were glabrous. No peduncle pubescence category was dominant among either core collection lines or cultivars. The high proportion of cultivars with glabrous petioles, stems and peduncles was largely attributable to the five ssp. *yanninicum* cultivars (S9 Table).

Ratings for extent of anthocyanin flecking and flushing of leaves and pigmentation in stipules and calyx tubes also showed wide diversity. Frequency distributions for both core collection lines and cultivars are shown in S5 Fig. Core collection lines featured in each rating category of these traits, apart from strong (6 rating) flecking and flushing, whereas the full range of trait expressions were only found among the cultivars for calyx pigmentation. For the leaf anthocyanin traits, 87.4% of core collection lines and 89.2% of cultivars had either nil or weak flecking (0–2 rating), while 86.1% of core collection lines and 89.3% of cultivars either lacked or had a weak flush pattern. No cultivars had a flecking rating > 3, while only one cultivar (Clare) had a flush rating > 3. Most core collection lines (74.2%) and cultivars (67. 8%) had moderately weak to moderately strong (3–5 rating) stipule pigmentation and all cultivars had some pigmentation. Most core collection lines (56.7%) and cultivars (50.0%) had no calyx

pigmentation. It was notable that the calyces of all ssp. *brachycalycinum* genotypes were not pigmented (S9 Table).

### Associations between plant traits

Correlations between the 23 quantitative agro-morphological and isoflavone traits for all 125 core collection lines and cultivars are shown in Table 3. Flowering time was correlated with several traits. Later flowering was associated with taller plants and greater leaf size at 110 DAS, longer internodes and higher biomass and maximum stem length at senescence. Flowering time was also negatively correlated with stem diameter and petiole length at first flowering and with mean seed weight. High genistein content was also associated with later flowering.

Clusters of inter-related traits were also observed (Table 3). For example, plant diameter, leaf size, petiole length and petiole diameter at 77 DAS were all highly correlated with each other. Similarly, plant size, biomass, plant height and leaf size at 110 DAS were all correlated.

At first flowering leaf size, petiole length and petiole diameter were highly correlated with each other (Table 3). These traits were also correlated with internode length and stem diameter, while the latter two traits were also weakly correlated. Long peduncles were associated with large leaves, long and thick petioles and long internodes at flowering and with long stems at senescence, but there was no association with peduncle diameter. Thick peduncles instead were associated with thick stems and with long, thick petioles at flowering. Seed size was associated with several traits and had the highest correlations with stem diameter, stem length and total biomass (Table 3).

In general, large leaf traits during the vegetative phase translated into large leaf traits at flowering (Table 3). For example, leaf size at 77 DAS, 110 DAS and at first flowering were all significantly correlated. This was also apparent for petiole length and petiole diameter. Several traits were good predictors of total seasonal biomass, with large plants at senescence being associated with large plant traits during the growing season. Leaf size, petiole length and petiole diameter measured as early as 77 DAS were able to predict total seasonal biomass, while traits measured at 110 DAS had even higher correlations.

Among the isoflavones, formononetin content was not correlated with any other plant trait, apart from total isoflavone content. (Table 3). High genistein content was associated with several plant traits, including large leaves and long, thick petioles and long peduncles. High Biochanin A content, on the other hand, was associated with small plants, short petioles and peduncles and small leaves. Of note was the strong negative correlation between genistein and biochanin A contents.

Correlations between morphological marker traits are shown in Table 4. Pubescence of stems, petioles and peduncles were inter-related. Leaf upper surface pubescence was associated with petiole pubescence, but not with stem or peduncle pubescence. Genotypes with strong stem pubescence also tended to have strong leaf anthocyanin flush patterns, strong calyx pigmentation and narrow leaf mark arms. Strong calyx pigmentation was also associated with large leaf mark crescents and narrow arms and weakly associated with strong leaf flecking and leaf anthocyanin flushing. Leaf flush intensity was also associated with crescent leaf mark size and degree of leaf flecking.

Somewhat surprisingly, significant correlations were found between several agro-morphological and morphological marker traits (Table 5). Among the plant growth traits, late flowering tended to be associated with leaf marks comprised of small crescents and broad arms, weak calyx tube pigmentation, weak upper leaf surface pubescence and a weak anthocyanin leaf flush. Long peduncles were associated with broad leaf mark arms, pubescent stems and strongly pigmented calyx tubes and stipules. High Biochanin A content was also associated

**Table 3. Pearson's correlation statistics for 23 agro-morphological traits among 97 core collection lines and 28 diverse cultivars.**

| Trait | N[a] | 1 | 2 | 3 | 4 | 5 | 6 | 7 | 8 | 9 | 10 | 11 | 12 | 13 | 14 | 15 | 16 | 17 | 18 | 19 | 20 | 21 | 22 |
|---|---|---|---|---|---|---|---|---|---|---|---|---|---|---|---|---|---|---|---|---|---|---|---|
| 1 Flowering time | 125 | 1 | | | | | | | | | | | | | | | | | | | | | |
| 2 Plant diameter at 77 days | 125 | 0.10 | | | | | | | | | | | | | | | | | | | | | |
| 3 Leaf size at 77 days | 125 | -0.01 | 0.85*** | | | | | | | | | | | | | | | | | | | | | |
| 4 Petiole diameter at 77 days | 125 | 0.03 | 0.84*** | 0.89*** | | | | | | | | | | | | | | | | | | | | |
| 5 Petiole length at 77 days | 125 | 0.20* | 0.93*** | 0.76*** | 0.76*** | | | | | | | | | | | | | | | | | | | |
| 6 Plant height at 110 days | 125 | 0.32*** | 0.54*** | 0.48*** | 0.41*** | 0.59*** | | | | | | | | | | | | | | | | | | |
| 7 Biomass at 110 days | 125 | -0.01 | 0.81*** | 0.77*** | 0.73*** | 0.76*** | 0.59*** | | | | | | | | | | | | | | | | | |
| 8 Plant area at 110 days | 125 | -0.15 | 0.80*** | 0.83*** | 0.79*** | 0.74*** | 0.47*** | 0.92*** | | | | | | | | | | | | | | | | |
| 9 Leaf size at 110 days | 125 | 0.29*** | 0.56*** | 0.46*** | 0.44*** | 0.54*** | 0.78*** | 0.64*** | 0.52*** | | | | | | | | | | | | | | | |
| 10 Leaf size at flowering | 125 | 0.17 | 0.53*** | 0.58*** | 0.52*** | 0.51*** | 0.55*** | 0.61*** | 0.53*** | 0.56*** | | | | | | | | | | | | | | |
| 11 Petiole length at flowering | 125 | -0.29*** | 0.18* | 0.16 | 0.11 | 0.16 | 0.25* | 0.33*** | 0.26** | 0.27* | 0.63*** | | | | | | | | | | | | | |
| 12 Petiole diameter at flowering | 125 | 0.07 | 0.50*** | 0.51*** | 0.51*** | 0.50*** | 0.47*** | 0.54*** | 0.49*** | 0.47*** | 0.87*** | 0.67*** | | | | | | | | | | | | |
| 13 Internode length | 125 | 0.21* | 0.51*** | 0.50*** | 0.48*** | 0.51*** | 0.33*** | 0.58*** | 0.53*** | 0.41*** | 0.76*** | 0.47*** | 0.63*** | | | | | | | | | | | |
| 14 Stem diameter | 125 | -0.43*** | 0.20* | 0.29** | 0.28** | 0.11 | 0.03 | 0.29** | 0.30** | 0.11 | 0.49*** | 0.63*** | 0.61*** | 0.25** | | | | | | | | | | |
| 15 Formononetin | 125 | -0.12 | 0.05 | -0.03 | -0.03 | -0.03 | -0.14 | -0.09 | -0.10 | 0.02 | -0.04 | 0.05 | 0.03 | 0.01 | 0.07 | | | | | | | | | |
| 16 Genistein | 125 | 0.25** | 0.31*** | 0.32*** | 0.34*** | 0.37*** | 0.35*** | 0.24*** | 0.22 | 0.31*** | 0.35*** | 0.04 | 0.30*** | 0.26** | -0.06 | -0.12 | | | | | | | | |
| 17 Biochanin A | 125 | 0.00 | -0.33*** | -0.26** | -0.29** | -0.38*** | -0.22* | -0.29** | -0.30*** | -0.17 | -0.17 | -0.02 | -0.19* | -0.19* | 0.10 | 0.14 | -0.55*** | | | | | | | |
| 18 Total isoflavones | 125 | 0.12 | -0.02 | 0.00 | 0.00 | -0.06 | -0.01 | -0.12 | -0.15 | 0.10 | 0.09 | 0.04 | 0.08 | 0.04 | 0.08 | 0.63*** | 0.21* | 0.53*** | | | | | | |
| 19 Max stem length | 124 | 0.34*** | 0.47*** | 0.48*** | 0.44*** | 0.39*** | 0.08 | 0.56*** | 0.58*** | 0.18* | 0.49*** | 0.44*** | 0.41*** | 0.58*** | 0.49*** | 0.12 | 0.06 | -0.10 | 0.03 | | | | | |
| 20 Mean seed weight | 124 | -0.44*** | 0.29** | 0.42*** | 0.35*** | 0.23** | 0.05 | 0.28** | 0.38*** | 0.02 | 0.33*** | 0.30*** | 0.36*** | 0.21* | 0.59*** | 0.19* | -0.12 | 0.12 | 0.13 | 0.50*** | | | | |
| 21 Peduncle length | 121 | 0.14 | 0.62*** | 0.58*** | 0.59*** | 0.59*** | 0.29** | 0.60*** | 0.62*** | 0.34*** | 0.53*** | 0.18* | 0.41*** | 0.64*** | 0.07 | -0.09 | 0.39*** | -0.44*** | -0.1 | 0.48*** | 0.09 | | | |
| 22 Total biomass | 117 | 0.25** | 0.50*** | 0.41*** | 0.39*** | 0.46*** | 0.45*** | 0.60*** | 0.45*** | 0.47*** | 0.60*** | 0.35*** | 0.46*** | 0.57*** | 0.13 | -0.12 | 0.23* | -0.20* | -0.1 | 0.59*** | 0.05 | 0.50*** | | |
| 23 Peduncle diameter | 114 | -0.09 | 0.01 | 0.05 | 0.01 | 0.03 | 0.10 | 0.06 | 0.00 | 0.03 | 0.21* | 0.32*** | 0.26** | 0.14 | 0.39*** | 0.08 | -0.16 | 0.25** | 0.13 | 0.18 | 0.39*** | -0.16 | 0.11 | |

[a] Number of genotypes measured

* $P \leq 0.05$

** $P \leq 0.01$

*** $P \leq 0.001$

Strength of positive correlations is shown by increasing intensities of blue; strength of negative correlations is shown by increasing intensities of brown

**Table 4. Pearson's correlation statistics for morphological marker traits among 97 core collection lines and 28 diverse cultivars.**

| | Trait | 1 | 2 | 3 | 4 | 5 | 6 | 7 | 8 | 9 | 10 |
|---|---|---|---|---|---|---|---|---|---|---|---|
| 1 | Leaflet indentation | | | | | | | | | | |
| 2 | Leaf mark: crescent width[a] | -0.03 | | | | | | | | | |
| 3 | Leaf mark: arm breadth | 0.01 | -0.10 | | | | | | | | |
| 4 | Leaf flecking | -0.02 | 0.15 | -0.27** | | | | | | | |
| 5 | Leaf flush | 0.14 | 0.44*** | -0.23* | 0.37*** | | | | | | |
| 6 | Stipule pigmentation | 0.13 | -0.09 | 0.26* | -0.23 | -0.09 | | | | | |
| 7 | Calyx pigmentation | -0.06 | 0.27** | -0.44*** | 0.25* | 0.23* | -0.19 | | | | |
| 8 | Leaf upper surface pubescence | -0.02 | -0.05 | 0.00 | -0.16 | -0.03 | 0.11 | 0.12 | | | |
| 9 | Petiole pubescence | -0.05 | -0.17 | 0.02 | -0.04 | 0.12 | 0.15 | -0.01 | 0.30** | | |
| 10 | Stem pubescence | 0.03 | 0.18 | -0.31** | 0.23 | 0.29** | -0.09 | 0.29** | 0.12 | 0.31** | |
| 11 | Peduncle pubescence | -0.06 | -0.06 | -0.20 | 0.02 | 0.16 | 0.00 | 0.14 | 0.20 | 0.57*** | 0.70*** |

[a]Bands were considered as modified $C_4$ crescents for this analysis

*$P \leq 0.05$

**$P \leq 0.01$

***$P \leq 0.001$

Strength of positive correlations is shown by increasing intensities of blue; strength of negative correlations is shown by increasing intensities of brown

with several traits, most notably a strong leaf anthocyanin flush and leaf flecking, pubescent stems and narrow leaf mark arms. Long internodes were also associated with several traits, particularly little or no calyx pigmentation and narrow leaf mark arms. Significant correlations with at least one morphological marker trait were found for all other agro-morphological traits, apart from petiole length at flowering.

Among the morphological marker traits, broad leaf mark arms and weak calyx pigmentation were both associated with high values for all plant growth traits measured at 77 DAS and 110 DAS, along with large leaves and thick petioles at flowering, long internodes and peduncles, large plants at maturity and high genistein content and with low biochanin A content (Table 5). Similar trends were also found between stem pubescence and most of these traits. In the case of calyx pigmentation, this data may be conflicted by subspecies differences, as no ssp. *brachycalycinum* genotypes had any calyx pigmentation (S9 Table) and they tended to have higher values than the other subspecies for these traits (Table 2). Such a subspecies influence was less apparent, however, for arm breadth and stem pubescence.

## Trait associations with sites of origin

Several agro-morphological and isoflavone traits were correlated with collection site parameters (Table 6). Flowering time was associated with latitude, precipitation in the summer months (BIO14, BIO17 and BIO18) and total annual precipitation (BIO12), but not with in-season precipitation (BIO13 and BIO16). The two strongest correlations (with latitude and BIO17) are illustrated in Fig 3. Later flowering was also associated with cooler mean annual temperature (BIO1) and cooler temperatures in both winter (BIO6, BIO9 and BIO11) and summer (BIO5 and BIO10). There was no relationship between flowering time and altitude.

Genotypes with more vigorous early winter growth, large leaves and thick petioles (measured at 77 DAS) tended to originate from lower latitude, warmer sites with shorter growing

**Table 5. Pearson's correlation statistics between agronomic and morphological traits among 97 core collection lines and 28 diverse cultivars.**

| Trait | Leaflet indentation | Leaf mark crescent width[a] | Leaf mark arm breadth | Leaf flecking | Leaf flush | Stipule pigment-ation | Calyx pigment-ation | Leaf upper surface pubescence | Petiole pubescence | Stem pubescence | Peduncle pubescence |
|---|---|---|---|---|---|---|---|---|---|---|---|
| Plant diameter at 77 days | -0.03 | -0.19* | 0.35*** | -0.11 | 0.04 | 0.15 | -0.42*** | -0.03 | 0.16 | -0.31*** | -0.04 |
| Leaf size at 77 days | -0.04 | -0.16 | 0.27** | -0.07 | 0.05 | 0.12 | -0.38*** | 0.02 | 0.15 | -0.27** | -0.03 |
| Petiole diameter at 77 days | -0.08 | -0.14 | 0.26** | -0.10 | 0.08 | 0.15 | -0.36*** | 0.02 | 0.16 | -0.30** | -0.02 |
| Petiole length at 77 days | -0.05 | -0.23* | 0.38*** | -0.12 | -0.01 | 0.16 | -0.48*** | -0.04 | 0.11 | -0.36*** | -0.06 |
| Plant height at 110 days | 0.07 | -0.10 | 0.29** | 0.00 | -0.11 | 0.17 | -0.33*** | -0.11 | 0.10 | -0.09 | 0.09 |
| Biomass at 110 days | 0.06 | -0.20* | 0.35*** | -0.05 | -0.04 | 0.12 | -0.39*** | 0.04 | 0.12 | -0.23* | 0.04 |
| Plant area at 110 days | 0.02 | -0.14 | 0.33*** | -0.07 | 0.02 | 0.15 | -0.35*** | 0.17 | 0.09 | -0.27** | -0.02 |
| Leaf size at 110 days | 0.20* | -0.20* | 0.26** | 0.01 | -0.15 | 0.14 | -0.38*** | -0.09 | 0.10 | -0.09 | 0.07 |
| Leaf size at flowering | 0.21* | -0.13 | 0.34*** | 0.03 | 0.00 | 0.12 | -0.45*** | -0.16 | 0.14 | -0.27** | -0.09 |
| Petiole length at flowering | 0.13 | 0.07 | 0.11 | 0.04 | 0.16 | 0.08 | -0.15 | -0.05 | 0.04 | -0.08 | 0.00 |
| Petiole diameter at flowering | 0.07 | -0.10 | 0.29** | -0.01 | 0.02 | 0.08 | -0.48*** | -0.14 | -0.01 | -0.36*** | -0.18 |
| Internode length | 0.21* | -0.26** | 0.40*** | -0.07 | 0.03 | 0.14 | -0.46*** | -0.08 | 0.13 | -0.28** | -0.05 |
| Stem diameter | 0.07 | 0.18 | -0.04 | 0.06 | 0.29** | -0.06 | -0.11 | -0.07 | 0.15 | 0.02 | 0.03 |
| Peduncle length | 0.12 | -0.22* | 0.52*** | -0.26** | -0.13 | 0.39*** | -0.43*** | 0.07 | 0.23* | -0.44*** | -0.12 |
| Peduncle diameter | 0.05 | 0.17 | -0.20* | 0.26** | 0.28** | -0.15 | 0.17 | -0.15 | 0.17 | 0.17 | 0.15 |
| Flowering time | 0.17 | -0.32*** | 0.26** | -0.04 | -0.30** | 0.04 | -0.34*** | -0.34*** | -0.01 | -0.18 | -0.11 |
| Max stem length | 0.06 | 0.04 | 0.21* | -0.05 | 0.24** | 0.02 | -0.20* | 0.06 | 0.18 | -0.15 | 0.06 |
| Total biomass | 0.11 | -0.19* | 0.33*** | -0.07 | -0.11 | 0.15 | -0.26** | -0.15 | 0.16 | -0.21* | 0.06 |
| Mean seed weight | 0.05 | 0.11 | -0.19* | 0.22* | 0.39*** | -0.16 | -0.08 | 0.04 | 0.16 | -0.02 | 0.07 |
| Formononetin | 0.05 | -0.05 | -0.04 | 0.05 | 0.25** | -0.13 | -0.05 | -0.06 | -0.06 | 0.04 | 0.05 |
| Genistein | 0.01 | -0.15 | 0.41*** | -0.18* | -0.14 | 0.05 | -0.28** | -0.16 | -0.01 | -0.22* | -0.14 |
| Biochanin A | 0.16 | 0.20* | -0.42*** | 0.35*** | 0.36*** | -0.15 | 0.20* | -0.24* | -0.06 | 0.30*** | 0.10 |
| Total isoflavones | 0.17 | 0.02 | -0.06 | 0.18 | 0.33*** | -0.16 | -0.08 | -0.35*** | -0.09 | 0.11 | 0.01 |

[a]Bands were considered as modified $C_4$ crescents for this analysis

$^*P \leq 0.05$

$^{**}P \leq 0.01$

$^{***}P \leq 0.001$

Strength of positive correlations is shown by increasing intensities of blue; strength of negative correlations is shown by increasing intensities of brown

seasons. Large leaves were also associated with high soil pH. Larger plants at the end of winter (110 DAS) tended to come from warmer sites. Plant diameter was also negatively correlated with latitude and precipitation in summer.

**Table 6. Pearson's correlation statistics between agro-morphological traits and site of origin data among the 94 subterranean clover core collection lines for which site of origin data was available.** See S3 Table for definitions of BIOCLIM variables.

| Trait | Passport data | | | | | BIOCLIM variables | | | | | | | | | | | | | | | | | | |
| | Latitude | Longitude | Altitude | Soil pH | Soil texture | BIO1 | BIO2 | BIO3 | BIO4 | BIO5 | BIO6 | BIO7 | BIO8 | BIO9 | BIO10 | BIO11 | BIO12 | BIO13 | BIO14 | BIO15 | BIO16 | BIO17 | BIO18 | BIO19 |
| | | | | | | Temperature (°C) | | | | | | | | | | | Precipitation (mm) | | | | | | | |
|---|---|---|---|---|---|---|---|---|---|---|---|---|---|---|---|---|---|---|---|---|---|---|---|---|
| No comparisons | 94 | 94 | 94 | 50 | 49 | 93 | 93 | 93 | 93 | 93 | 93 | 93 | 93 | 93 | 93 | 93 | 93 | 93 | 93 | 93 | 93 | 93 | 93 | 93 |
| Flowering time | 0.53*** | 0.01 | -0.05 | 0.01 | 0.02 | -0.35*** | -0.11 | -0.14 | 0.05 | -0.27** | -0.29** | -0.03 | -0.18 | -0.28** | -0.33** | -0.31** | 0.27** | 0.01 | 0.49*** | -0.41*** | 0.04 | 0.51*** | 0.49*** | 0.00 |
| Plant diameter at 77 days | -0.16 | 0.01 | -0.10 | 0.33* | -0.03 | 0.20 | -0.07 | 0.04 | -0.11 | 0.05 | 0.23* | -0.12 | 0.07 | 0.15 | 0.15 | 0.20 | -0.09 | 0.00 | -0.19 | 0.21* | 0.00 | -0.21* | -0.21* | -0.01 |
| Leaf size at 77 days | -0.36*** | 0.03 | -0.16 | 0.33* | -0.05 | 0.35*** | 0.06 | 0.19 | -0.11 | 0.17 | 0.30** | -0.06 | 0.13 | 0.23* | 0.29** | 0.32** | -0.13 | 0.04 | -0.23* | 0.33** | 0.02 | -0.28** | -0.27** | 0.00 |
| Petiole diameter at 77 days | -0.32** | 0.15 | -0.19 | 0.38** | -0.01 | 0.28** | -0.03 | 0.12 | -0.11 | 0.07 | 0.27** | -0.12 | 0.08 | 0.21* | 0.21* | 0.26* | -0.04 | 0.10 | -0.16 | 0.30** | 0.09 | -0.22* | -0.21* | 0.09 |
| Petiole length at 77 days | -0.11 | 0.02 | -0.08 | 0.29* | 0.01 | 0.13 | -0.09 | 0.02 | -0.09 | 0.00 | 0.18 | -0.12 | 0.04 | 0.13 | 0.08 | 0.14 | -0.07 | -0.01 | -0.16 | 0.16 | -0.01 | -0.17 | -0.19 | 0.00 |
| Plant height at 110 days | 0.08 | 0.01 | 0.06 | 0.08 | -0.02 | -0.07 | -0.08 | -0.08 | 0.04 | -0.07 | -0.04 | -0.03 | -0.05 | -0.08 | -0.07 | -0.08 | 0.00 | -0.02 | 0.06 | -0.02 | -0.02 | 0.06 | 0.03 | -0.02 |
| Biomass at 110 days | -0.19 | -0.02 | -0.16 | 0.12 | -0.06 | 0.25* | -0.02 | 0.15 | -0.17 | 0.07 | 0.27** | -0.13 | 0.15 | 0.15 | 0.16 | 0.26* | -0.12 | -0.02 | -0.16 | 0.22* | -0.03 | -0.18 | -0.20 | -0.05 |
| Plant area at 110 days | -0.33** | -0.01 | -0.16 | 0.22 | -0.06 | 0.30** | 0.05 | 0.22* | -0.16 | 0.13 | 0.30** | -0.10 | 0.13 | 0.19 | 0.21* | 0.29** | -0.17 | -0.02 | -0.24* | 0.30** | -0.04 | -0.27** | -0.30** | -0.03 |
| Leaf size at 110 days | 0.04 | -0.05 | 0.07 | -0.02 | -0.02 | -0.01 | -0.05 | -0.03 | -0.02 | -0.04 | 0.01 | -0.04 | 0.02 | -0.03 | -0.03 | -0.01 | 0.01 | -0.01 | 0.05 | -0.01 | -0.01 | 0.05 | 0.04 | -0.04 |
| Leaf size at flowering | -0.02 | 0.06 | -0.25* | 0.01 | 0.02 | 0.15 | 0.08 | 0.08 | 0.04 | 0.12 | 0.05 | 0.07 | 0.18 | -0.06 | 0.15 | 0.09 | -0.02 | -0.01 | -0.03 | 0.05 | -0.01 | -0.03 | -0.05 | -0.05 |
| Petiole length at flowering | -0.04 | 0.02 | -0.12 | -0.34* | 0.04 | 0.18 | 0.07 | 0.08 | -0.02 | 0.13 | 0.15 | 0.04 | 0.29** | -0.07 | 0.17 | 0.14 | -0.15 | -0.07 | -0.15 | 0.08 | -0.08 | -0.14 | -0.12 | -0.14 |
| Petiole diam. at flowering | -0.07 | 0.09 | -0.24* | -0.02 | 0.04 | 0.21* | 0.04 | 0.13 | -0.07 | 0.08 | 0.14 | -0.03 | 0.21 | 0.01 | 0.16 | 0.18 | -0.03 | 0.05 | -0.07 | 0.15 | 0.04 | -0.08 | -0.08 | -0.03 |
| Internode length | 0.07 | 0.07 | -0.36*** | 0.17 | 0.13 | 0.14 | 0.08 | 0.11 | 0.03 | 0.12 | 0.04 | 0.07 | 0.16 | -0.04 | 0.14 | 0.09 | -0.09 | -0.08 | -0.04 | 0.01 | -0.08 | -0.04 | -0.07 | -0.09 |
| Stem diameter | -0.26* | 0.04 | -0.19 | -0.13 | -0.14 | 0.23* | 0.07 | 0.15 | -0.08 | 0.12 | 0.17 | -0.01 | 0.19 | 0.07 | 0.18 | 0.20 | -0.10 | 0.01 | -0.15 | 0.16 | 0.00 | -0.17 | -0.18 | 0.00 |
| Peduncle length | -0.04 | 0.06 | -0.24* | 0.43** | 0.31* | 0.12 | -0.08 | -0.02 | -0.03 | 0.04 | 0.16 | -0.08 | 0.13 | 0.08 | 0.11 | 0.12 | -0.22* | -0.18 | -0.17 | 0.04 | -0.19 | -0.19 | -0.15 | -0.16 |
| Peduncle diameter | 0.07 | -0.13 | 0.03 | -0.20 | -0.04 | -0.04 | 0.00 | 0.04 | -0.11 | -0.08 | -0.02 | -0.05 | 0.03 | -0.13 | -0.10 | -0.01 | 0.17 | 0.08 | 0.20 | -0.09 | 0.10 | 0.20 | 0.23* | 0.02 |
| Max stem length | -0.18 | -0.03 | -0.26* | 0.17 | 0.02 | 0.32** | 0.16 | 0.19 | -0.01 | 0.28** | 0.20 | 0.10 | 0.21* | 0.18 | 0.31** | 0.25** | -0.21* | -0.10 | -0.29** | 0.19 | -0.10 | -0.29** | -0.32** | -0.07 |
| Total biomass | 0.16 | -0.04 | -0.12 | 0.07 | 0.10 | 0.10 | -0.12 | -0.03 | -0.14 | -0.05 | 0.15 | -0.14 | 0.25* | -0.06 | 0.03 | 0.12 | -0.02 | -0.08 | 0.02 | -0.04 | -0.05 | 0.02 | 0.09 | -0.11 |
| Mean seed weight | -0.38*** | -0.06 | -0.14 | 0.18 | -0.08 | 0.21* | 0.14 | 0.19 | -0.04 | 0.16 | 0.12 | 0.05 | 0.06 | 0.13 | 0.18 | 0.17 | -0.06 | 0.02 | -0.16 | 0.19 | 0.02 | -0.18 | -0.23* | 0.06 |
| Formononetin | -0.04 | 0.04 | -0.11 | 0.03 | -0.14 | 0.06 | 0.28** | 0.10 | 0.28** | 0.28** | -0.16 | 0.34*** | -0.21* | 0.16 | 0.19 | -0.06 | 0.09 | 0.06 | 0.02 | -0.04 | 0.07 | 0.03 | -0.04 | 0.13 |
| Genistein | 0.09 | 0.02 | 0.01 | 0.06 | 0.13 | -0.07 | -0.18 | -0.23* | 0.06 | -0.05 | 0.05 | -0.08 | 0.04 | -0.02 | -0.03 | -0.05 | -0.04 | -0.10 | -0.17 | -0.07 | -0.08 | -0.13 | -0.08 | -0.04 |
| Biochanin A | 0.13 | -0.28** | -0.13 | -0.40** | -0.14 | -0.04 | 0.26* | 0.28** | -0.03 | 0.04 | -0.18 | 0.15 | -0.06 | -0.15 | -0.07 | -0.06 | 0.07 | -0.07 | 0.38*** | -0.28** | -0.07 | 0.39*** | 0.33 | -0.09 |
| Total isoflavones | 0.17 | -0.22* | -0.18 | -0.22 | -0.08 | -0.05 | 0.26* | 0.15 | 0.18 | 0.16 | -0.22* | 0.29** | -0.14 | -0.06 | 0.02 | -0.13 | 0.09 | -0.10 | 0.23* | -0.34** | -0.08 | 0.28** | 0.22 | -0.04 |

* $P \leq 0.05$

** $P \leq 0.01$

*** $P \leq 0.001$

Strength of positive correlations is shown by increasing intensities of blue; strength of negative correlations is shown by increasing intensities of brown

**Table 7. Pearson's correlation statistics between morphological marker traits and site of origin data among the 94 subterranean clover core collection lines for which site of origin data was available.** See S3 Table for definitions of BIOCLIM variables.

| Trait | Passport data from the ATGRC | | | | | BIOCLIM variables | | | | | | | | | | | | | | | | | | |
|---|---|---|---|---|---|---|---|---|---|---|---|---|---|---|---|---|---|---|---|---|---|---|---|---|
| | | | | | | Temperature (°C) | | | | | | | | | | | Precipitation (mm) | | | | | | | |
| | Altitude | Latitude | Longitude | Soil pH | Soil texture | BIO1 | BIO2 | BIO3 | BIO4 | BIO5 | BIO6 | BIO7 | BIO8 | BIO9 | BIO10 | BIO11 | BIO12 | BIO13 | BIO14 | BIO15 | BIO16 | BIO17 | BIO18 | BIO19 |
| No comparisons | 94 | 94 | 94 | 50 | 49 | 93 | 93 | 93 | 93 | 93 | 93 | 93 | 93 | 93 | 93 | 93 | 93 | 93 | 93 | 93 | 93 | 93 | 93 | 93 |
| Leaflet indentation | 0.01 | 0.07 | -0.16 | -0.15 | -0.08 | -0.06 | 0.07 | 0.14 | -0.11 | -0.05 | -0.04 | -0.01 | -0.03 | -0.22* | -0.11 | -0.02 | 0.00 | -0.02 | -0.01 | -0.03 | -0.03 | 0.02 | 0.03 | -0.02 |
| Leaf mark crescent size | -0.09 | -0.03 | -0.12 | -0.12 | -0.03 | 0.16 | 0.02 | 0.05 | -0.07 | 0.09 | 0.17 | -0.04 | 0.18 | 0.03 | 0.14 | 0.16 | -0.16 | -0.14 | -0.06 | -0.03 | -0.16 | -0.04 | 0.01 | -0.15 |
| Leaf mark arm breadth | -0.21 | 0.18 | 0.02 | 0.33** | 0.10 | 0.11 | 0.06 | 0.09 | -0.02 | 0.10 | 0.08 | 0.03 | 0.15 | 0.08 | 0.11 | 0.10 | -0.10 | -0.09 | -0.07 | 0.01 | -0.10 | -0.05 | -0.03 | -0.11 |
| Leaf flecking | -0.10 | 0.14 | -0.36*** | -0.33** | -0.24 | 0.02 | 0.01 | 0.07 | -0.13 | -0.03 | 0.06 | -0.06 | 0.11 | 0.00 | -0.03 | 0.05 | -0.04 | -0.18 | 0.17 | -0.28** | -0.17 | 0.22* | 0.24* | -0.15 |
| Leaf flush | -0.11 | -0.03 | -0.28** | -0.30* | -0.17 | 0.13 | 0.16 | 0.17 | -0.03 | 0.18 | 0.08 | 0.09 | 0.07 | 0.06 | 0.13 | 0.11 | -0.06 | -0.14 | 0.02 | -0.12 | -0.13 | 0.04 | 0.04 | -0.10 |
| Stipule pigmentation | -0.02 | -0.02 | -0.05 | 0.30* | 0.25* | 0.12 | 0.09 | 0.12 | -0.07 | 0.11 | 0.09 | 0.03 | 0.15 | -0.05 | 0.11 | 0.14 | -0.15 | -0.10 | -0.15 | 0.08 | -0.11 | -0.18 | -0.05 | -0.14 |
| Calyx pigmentation | 0.12 | -0.06 | -0.01 | -0.28* | -0.15 | -0.02 | -0.15 | -0.13 | -0.08 | -0.09 | 0.10 | -0.15 | 0.05 | 0.02 | -0.05 | 0.02 | -0.03 | -0.02 | -0.08 | -0.02 | 0.00 | -0.06 | -0.02 | 0.02 |
| Leaf pubescence | 0.07 | -0.23* | 0.31** | 0.07 | -0.22 | 0.06 | -0.01 | -0.06 | 0.13 | 0.06 | 0.01 | 0.04 | -0.06 | 0.11 | 0.10 | 0.00 | -0.09 | 0.10 | -0.18 | 0.28** | 0.07 | -0.21* | -0.26* | 0.14 |
| Petiole pubescence | 0.09 | 0.00 | 0.04 | 0.13 | 0.17 | -0.11 | -0.11 | -0.14 | 0.04 | -0.11 | -0.07 | -0.05 | -0.02 | -0.13 | -0.10 | -0.10 | 0.06 | -0.02 | 0.12 | -0.12 | -0.01 | 0.08 | 0.09 | 0.03 |
| Stem pubescence | 0.16 | 0.09 | -0.05 | -0.25* | -0.23 | -0.13 | -0.04 | -0.11 | 0.07 | -0.06 | -0.12 | 0.03 | -0.07 | -0.09 | -0.09 | -0.13 | 0.09 | -0.04 | 0.24* | -0.22* | -0.04 | 0.25* | 0.25* | -0.02 |
| Peduncle pubescence | 0.08 | 0.04 | 0.12 | -0.08 | -0.12 | -0.16 | -0.06 | -0.15 | 0.16 | -0.10 | -0.18 | 0.04 | -0.16 | -0.05 | -0.11 | -0.20 | 0.13 | 0.00 | 0.28** | -0.20 | 0.02 | 0.26* | 0.21* | 0.09 |

\* $P \leq 0.05$

\*\* $P \leq 0.01$

\*\*\* $P \leq 0.001$

Strength of positive correlations is shown by increasing intensities of blue; strength of negative correlations is shown by increasing intensities of brown

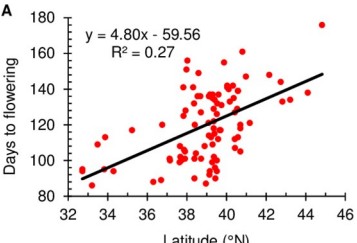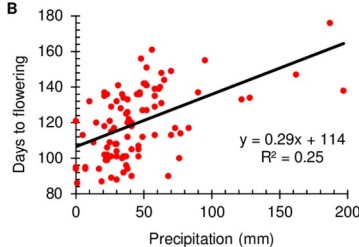

**Fig 3. Scatter plots of days to first flowering at Shenton Park, Western Australia, of subterranean clover core collection lines with known latitude and longitude of their collection sites against the two collection site variables with highest correlation.** (A) Latitude (°N). (B) BIOCLIM variable BIO17 (precipitation of the driest quarter).

There were few relationships between plant traits at first flowering and climatic variables, with the strongest being petiole length with mean summer temperature (BIO8). However, there were stronger relationships between traits at flowering and other passport data. In particular, internode length had a strong negative correlation with altitude, and peduncle length was positively associated with soil pH. It should be noted, however, that peduncle length associations with soil type may be due to the influence of subspecies, with ssp. *brachycalycinum* having longer peduncles (Table 2) and being derived from soils with higher mean pH and heavier texture (S1 Table).

At plant maturity total biomass was not strongly associated with any site character. Genotypes with longer stems tended to come from lower altitudes with warmer climates and lower summer precipitation. Genotypes with larger seeds tended to originate from lower latitudes with higher mean annual temperature and lower summer precipitation.

High formononetin content was related to sites with high temperature fluctuations, both seasonally (BIO4 and BIO6) and diurnally (BIO2), and with high maximum summer temperatures. High Biochanin A content was strongly associated with low soil pH, largely due to a high proportion of ssp. *brachycalycinum* (Table 2), high summer precipitation and low isothermality (BIO3), this being the difference in maximum summer and minimum winter temperatures. Biochanin A content also decreased with increasing longitude. Genistein content was not strongly related to any site attribute. Total isoflavone content had the strongest positive correlations with minimum temperature of the coldest month (BIO6) and precipitation over summer (BIO17).

Several morphological marker traits were correlated with collection site parameters (Table 7). Among the stronger associations, leaf mark arm breadth was correlated with soil pH, while strong leaf flecking and flush patterns tended to come from more westerly sites with more acidic soils. Genotypes with strong leaf upper surface pubescence tended to originate from more easterly and lower latitude sites with greater seasonality of precipitation (BIO15), while peduncle pubescence was associated with higher summer precipitation.

## Discussion

### Use of a core collection to measure subterranean clover diversity

This study confirmed subterranean clover is morphologically highly diverse. Ghamkhar et al. [4] previously demonstrated diversity across the species for flowering time and content of the isoflavones, formononetin, genistein and biochanin A and for total isoflavone content. Several studies have also examined diversity for flowering time within localised collections [17, 18,19, 20, 39], while Piano et al. [18, 19] and Pecetti and Piano [20,21] also examined diversity for leaf size and seed size from collections made in Sardinia and Sicily, Italy. This study, however,

reports on diversity for a further 16 quantitative agro-morphological traits across the known species range. Gladstones [15] examined morphological marker traits within a collection of naturalised strains in Western Australia, but this is the first study of diversity for these traits across the species breadth and no attempts have previously been made to relate them to their sites of origin.

This study demonstrated the value in using a well-defined core collection to measure diversity within subterranean clover. Clearly, it would not have been possible within a reasonable budget to examine a breadth of traits across the global collection of around 10,000 lines. The subterranean clover core collection has also been used to identify diversity for other traits, including methanogenic potential in the rumen [30], percentage of root length colonised by arbuscular mycorrhizal fungi [40] and cotyledon resistance to redlegged earth mite, *Halotydeus destructor*, [41].

Although the core collection encompassed a wide range of trait expression in this study, it was not as wide as that reported by Ghamkhar et al. [4] for the traits measured in the global collection of subterranean clover. The range of flowering times among the core collection (86–176 days from sowing) compares with 62–196 days for the global collection [4]. The maximum values for formononetin, genistein and biochanin A were also lower than those reported by Ghamkhar et al. [4]. This is to be expected, as the core collection only represents ~80% of the diversity within subterranean clover [7, 30]. However, the data from Ghamkhar et al. [4] also needs to be treated with caution, as it was sourced from unreplicated plots grown at different times over a 60-year period at three sites within the Perth metropolitan area and included data collected in Spain. By contrast, the data in this study was obtained from a single replicated trial grown at a common site. For the 13 morphological marker traits the core collection spanned all rating categories of Ghamkhar et al. [4], apart from 'very strong' ratings for leaf and petiole pubescence and 'strong' ratings for anthocyanin leaf flecking and flushing.

The hypothesis that there is similar diversity to the core collection among the Australian cultivars of subterranean clover was mostly supported. Of 23 agro-morphological and isoflavone traits, significant differences between means of the core collection and 28 test cultivars were found for only five traits. Flowering time of the cultivars was earlier on average than the core collection and there was a lower proportion of late flowering genotypes. This reflects the emphasis on selection of early flowering cultivars for low rainfall environments in Australian breeding programs [7]. Greater petiole length and diameter at flowering among the cultivars also reflects selection for high spring biomass but is also biased by high mean values of the five ssp. *yanninicum* cultivars. Stem length at plant maturity was also greater among the cultivars. As there were no differences for internode length between the core lines and cultivars, it suggests cultivars have more nodes along each stem, particularly flowering nodes, which may be the result of selection for high seed production [7]. There was also considerable diversity among the cultivars for morphological marker traits, although they had fewer trait expression categories than the core collection.

The finding that the core collection did not differ from the cultivars for formononetin content was surprising, given the heavy selection pressure for low levels of this isoflavone in breeding programs since the 1960s [7, 25, 26, 27]. The mean formononetin level of 0.25% was similar to that measured in Ghamkhar et al. [4]. However, only 7% of core collection lines were highly oestrogenic (>1.0% of d.m.), while 72% had levels < 0.2% of d.m., a level considered 'safe' for grazing [7, 27, 28]. These results support the assertion of Nichols et al. [16] that high formononetin content is not important for ecological success in the wild.

## Subspecies differences

Among the 97 core collection lines, mean flowering time of ssp. *brachycalycinum* was more than two weeks later than ssp. *subterraneum*, similar to the results of Piano et al. [18]. However, the range in flowering time was greater in ssp. *subterraneum*, in contrast with Ghamkhar et al. [4], who found a wider flowering time range in ssp. *brachycalycinum* in the world collection of subterranean clover. This may reflect the smaller sample size in this study. Across all 125 lines, the finding that ssp. *brachycalycinum* produced larger plants with larger leaves and longer internodes than the other two subspecies, while ssp. *yanninicum* had longer, thicker petioles, was also shown by Pecetti et al. [21] and Baresel et al. [42]. The plant architecture of ssp. *brachycalycinum* in the reproductive phase also differed from the other two subspecies by having thinner stems with longer internodes and longer, thinner peduncles, consistent with previous reports on subspecies characteristics [2, 7, 10]. The finding that ssp. *brachycalycinum* had lower formononetin and biochanin A contents and lower total isoflavone levels than the other two subspecies, while ssp. *subterraneum* had lower genistein content, is consistent with the observations of Nichols et al. [7].

## Plant trait associations

The relationships between flowering time and other key agro-morphological traits, particularly petiole length, leaf size and hardseededness (seed coat dormancy), are widely recognised for their importance to adaptation and agronomic performance [7, 16, 17, 18, 19, 20]. However, this study also found relationships between leaf marks, pubescence and anthocyanin traits with several agro-morphological traits. The significance of this finding is unclear. Part of this is attributable to the confounding of subspecies on trait expression. For example, no ssp. *brachycalycinum* genotypes had any calyx pigmentation (S9 Table) and this subspecies tended to have higher values for plant growth traits than the other subspecies (Table 2). Subspecies differences for agronomic traits also accounted for many of the observed associations with stem pubescence and leaf mark arm breadth.

Among the morphological traits, pubescence of stems, peduncles, petioles and leaf upper surfaces were all associated with each other. This suggests some common or linked genes may be involved in expression of pubescence across different plant organs. However, the relatively weak correlations in most cases suggest several genes are likely to be involved. The most common leaf marking of a crescent with arms was also found by Pecetti and Piano [20]. Several studies have shown the leaf mark trait is highly heritable, controlled by a single allele at a locus with >30 different alleles [31, 43]. The association between calyx pigmentation and leaf marks found in this study has also been previously shown [31].

## Relationships between plant traits and collection site parameters

This study supports the hypothesis that some specific plant traits are associated with environmental factors at their site of origin and are hence adaptive. It confirms previous annual legume studies that flowering time is highly responsive to latitude, temperature and precipitation (and their inter-relationships) in subterranean clover [17, 18, 19, 20, 39], purple clover (*T. purpureum* L.) [44], lentils (*Lens culinaris* Medik.) [45] and a range of Syrian annual legumes [46]. Early flowering is required in low rainfall, short-season areas to enable set seed before the onset of the long summer drought period, whereas later flowering enables a longer period of vegetative growth to better compete with other pasture components before flowering [16, 47].

Genotypes with more vigorous winter growth tended to originate from lower latitude, warmer sites with shorter growing seasons. This is consistent with Pecetti and Piano [21], who found earlier flowering genotypes tended to have larger leaves at 60 DAS among a range of

Sardinian subterranean clover ecotypes. Genotypes from high altitudes also tended to have shorter internodes and smaller leaves, similar to previous studies [21, 42], while maximum stem length also decreased with altitude. This suggests that plants from more elevated environments tend to have more compact growth, possibly to reduce the impacts of tissue damage from frost and extreme cold. The association of total biomass at plant maturity with mean summer temperature supports Baresel et al. [42], who found that plants from warmer climates produced more biomass. Among the soil characters, the increase in peduncle length with increasing soil pH is confounded by the preference of ssp. *brachycalycinum* for more neutral to alkaline soils than the other two subspecies [2, 4, 14]. Similarly, the association between low biochanin A content and high soil pH is largely attributable to ssp. *brachycalycinum*.

Among the agro-morphological traits, time to flowering, leaf size and petiole diameter at 77 DAS, plant area at 110 DAS, maximum stem length,and biochanin A and total isoflavone contents were correlated with seven or more environmental variables. These can be considered highly adaptive, being the result of strong environmental selection pressure over time. The importance of selection for this group of traits for adaptation to climate change is clear. By contrast, several traits had two or less significant correlations with environmental variables and can be considered adaptively neutral. This includes petiole length and plant height at 77 DAS, leaf size at 110 DAS and at flowering, petiole length at flowering, internode length, stem and peduncle diameter, total biomass and genistein content.

Unlike agro-morphological traits, which could be expected to have some adaptive relationships, morphological marker traits, commonly used to distinguish subterranean clover cultivars [7, 28], were not *a priori* expected to bear such relationships. However, some significant relationships with rainfall and soil parameters were found. This is the first time that morphological marker traits in a clover species have been linked to environmental adaptation. Although Pecetti and Piano [20] found differences in leaf mark types and calyx tube pigmentation between Sardinian populations, their sites of origin were climatically similar and no attempts were made to associate differences with climatic or edaphic factors. The reasons and significance of these associations are not clear, but may be related to genes for morphological marker traits being linked to other genes controlling adaptive traits that are responsive to different environments. Epigenetic inheritance may also be important [48] and needs further investigation.

## Plant breeding implications

The high genetic diversity and high heritability of the traits in this study indicates that selection gains can be readily made for each trait under similar conditions to this study. Indeed, many of these traits, particularly flowering time, isoflavone content and winter and spring biomass, are important selection criteria in current subterranean clover breeding programs [7]. It should be noted, however, that whereas all measurements were conducted in a common garden, trait expression could differ at different locations and at their sites of origin.

The 97-member core collection and the 28 diverse cultivars used in this study have recently been genotyped to develop a high-density single nucleotide polymorphism (SNP) linkage map [30]. Genome-wide association (GWAS) mapping has already been conducted for enteric methane emission and candidate genes have been identified [30]. Future GWAS mapping will be conducted on the traits measured in this study, with the prospect of molecular markers being identified. GWAS can also be applied to other traits that have been phenotyped using the core collection. The identification of polymorphic DNA-based molecular markers presents the opportunity for marker-assisted selection (MAS), which can be used to select among

crossbred progeny for closely linked target traits, particularly where conventional phenotyping methods are difficult or unreliable [7].

This study demonstrated that core collections are a useful tool for examining trait diversity to determine whether plant breeding solutions are feasible. However, the main weakness of core collections is that rare adaptive alleles may not be present among the core collection members [29, 49]. An alternative approach is the Focused Identification of Germplasm Strategy (FIGS), which uses both trait and climate data to develop *a priori* information to define a set of accessions with a high probability of containing desired traits [50]. This approach has been used to identify traits among several diverse collections, including drought adaptation in *Vicia faba* [49] and resistance to both powdery mildew (*Blumeria graminis* (DC) Speer f.sp. *tritici*) [51] and Russian wheat aphid (*Diuraphis noxia* Kurdj.) [52] in wheat (*Triticum aestivum*). This approach could be used on the subterranean clover global collection to identify new sources of diversity not present in the core collection.

## Supporting information

**S1 Fig. Mean monthly maximum and minimum temperatures in 2014 for Swanbourne, Western Australia, 3 km from the characterisation site (Data from the Australian Bureau of Meteorology).**
(TIF)

**S2 Fig. Frequency distributions of 97 subterranean clover core collection lines and 28 diverse Australian cultivars for leaf mark types, consisting of: (i) no leaf mark (nil); (ii) a pale green triangular crescent only; (iii) white or pale green arms only, extending from the margins towards the leaflet centre; (iv) a crescent and arms; or (v) a band, consisting of a pale green stripe extending from margin to margin.** Rating categories are given in S2 Table.
(TIF)

**S3 Fig. Frequency distributions of 97 subterranean clover core collection lines and 28 diverse Australian cultivars for leaflet indentation rating of the distal margin, ranging from 0 (no indentation) to 4 (strong indentation).** Rating categories are given in S2 Table.
(TIF)

**S4 Fig. Frequency distributions of pubescence ratings among 97 subterranean clover core collection lines and 28 diverse Australian cultivars, where 0 = glabrous, 2 = weak, 4 = moderate, 6 = strong, and 8 = very strong.** (A) Leaf upper surfaces. (B) Petioles. (C) Stems. (D) Peduncles. Ratings for leaf upper surfaces and petioles were conducted on September 5, while ratings for stems and peduncles were made two weeks after the commencement of flowering. Rating categories are given in S2 Table.
(TIF)

**S5 Fig. Frequency distributions of anthocyanin ratings among 97 subterranean clover core collection lines and 28 diverse Australian cultivars, where 0 = absent, 2 = weak, 4 = moderate, 6 = strong, and 8 = very strong.** (A) Leaf flecks. (B) Leaf flush. (C) Stipules. (D) Calyx tubes. Ratings for leaf flecks, leaf flush and stipule anthocyanin were conducted on September 5, while ratings for calyx tube pigmentation were made two weeks after the commencement of flowering. Rating categories are given in S2 Table.
(TIF)

**S1 Table. Subspecies, country of origin of the original wild accessions and passport data (where available) for the 97 subterranean clover core collection lines, consisting of 95 lines collected from the Mediterranean region and two crossbred cultivars (L96 and L97).** Data

from the Australian Trifolium Genetic Resource Centre (ATGRC), Department of Primary Industries and Regional Development. Collection sites from islands are shown in parentheses. Blanks indicate missing data.
(XLSX)

**S2 Table. Semi-quantitative morphological markers and the rating scales used for their expression (from Nichols et al. [28]).** Intermediate ratings were also used for leaflet indentation, leaf marks, anthocyanin intensity and pubescence characters, where applicable.
(XLSX)

**S3 Table. Climatic variables from BIOCLIM [36, 37] used in trait correlations for the 94 accessions of the core collection with known location.**
(XLSX)

**S4 Table. Transformations used for Analysis of Variance (ANOVA) to compare means of the 125 core collection lines and cultivars and to compare differences between subspecies *subterraneum*, *yanninicum* and *brachycalycinum* for 23 traits.** Residuals were analysed for assumptions of normality and homogeneity and square root or log transformations were conducted where appropriate when these assumptions were not fulfilled.
(XLSX)

**S5 Table. Data for 19 BIOCLIM variables for 93 core collection lines.** L12 and L79 are excluded as BIOCLIM data could not be obtained for their collection sites. L96 and L97 are excluded, as they are cross-bred cultivars. Explanations of variables are given in S3 Table.
(XLSX)

**S6 Table. Measurements at 77 days after sowing (DAS) and 110 DAS on single plants of 97 core collection lines and 28 diverse cultivars of subterranean clover.** Mean of four single plant replicates.
(XLSX)

**S7 Table. Days from sowing to flowering, measurements on the first flowering node and mature plant measurements on single plants of 97 core collection lines and 28 diverse cultivars of subterranean clover.** Mean of four singe plant replicates.
(XLSX)

**S8 Table. Oestrogenic isoflavone contents (% of dry matter) of 97 core collection lines and 28 diverse cultivars of subterranean clover.** Mean of four single plant replicates.
(XLSX)

**S9 Table. Morphological marker trait ratings on spaced plants of the 97 core collection lines and 28 diverse cultivars.** Ratings were conducted on single plants in the first two replicates and checked against plants in the third replicate. Rating scales for all traits are described in S2 Table.
(XLSX)

## Acknowledgments

Passport data were provided by Richard Snowball (DPIRD). Rosemarie Lugg conducted the isoflavone analyses. Special thanks are also given to Andrew Van Burgel (DPIRD) for statistical advice and to Ashty Saleem (Curtin University) who helped with the biomass ratings. AA conducted this research as part of his MSc studies, funded by the Kurdistan Regional Government under the KRG-Scholarship program, Human Capacity Development (HCDP). Funding was

also provided by Meat & Livestock Australia, as part of the project 'Pre-breeding in annual legumes' (P.PBE.0037). PN and BW were employed by DPIRD at the time this research was conducted.

## Author Contributions

**Conceptualization:** Phillip G. H. Nichols.

**Data curation:** Abdi I. Abdi, Parwinder Kaur, Bradley J. Wintle.

**Formal analysis:** Abdi I. Abdi, William Erskine.

**Funding acquisition:** Phillip G. H. Nichols.

**Investigation:** Abdi I. Abdi, Bradley J. Wintle.

**Methodology:** Abdi I. Abdi, Phillip G. H. Nichols, Bradley J. Wintle.

**Project administration:** Phillip G. H. Nichols.

**Supervision:** Phillip G. H. Nichols, Parwinder Kaur, William Erskine.

**Writing – original draft:** Abdi I. Abdi.

**Writing – review & editing:** Phillip G. H. Nichols, Parwinder Kaur, William Erskine.

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
