## [Decision Letter · Decision Letter 0]

19 Nov 2019

PONE-D-19-26944

Morphological diversity within a core collection of subterranean clover (Trifolium subterraneum L.): lessons in pasture adaptation from the wild

PLOS ONE

Dear Dr Nichols,

Thank you for submitting your manuscript to PLOS ONE. After careful consideration, we feel that it has merit but does not fully meet PLOS ONE’s publication criteria as it currently stands. Therefore, we invite you to submit a revised version of the manuscript that addresses the points raised during the review process.

The reviewers were genrally satisfied by the work. They suggest some detailed modifications to imporve the manuscript. Please consider them in your revised version.

We would appreciate receiving your revised manuscript by Jan 03 2020 11:59PM. To enhance the reproducibility of your results, we recommend that if applicable you deposit your laboratory protocols in protocols.io, where a protocol can be assigned its own identifier (DOI) such that it can be cited independently in the future. For instructions see: http://journals.plos.org/plosone/s/submission-guidelines#loc-laboratory-protocols

We look forward to receiving your revised manuscript.

Kind regards,

Sergio Rossi

Academic Editor

PLOS ONE

Journal Requirements:

Reviewers' comments:

Reviewer's Responses to Questions

**Comments to the Author**

1. Is the manuscript technically sound, and do the data support the conclusions?

Reviewer #1: Yes

Reviewer #2: Yes

Reviewer #3: Yes

2. Has the statistical analysis been performed appropriately and rigorously? 

Reviewer #1: Yes

Reviewer #2: Yes

Reviewer #3: Yes

3. Have the authors made all data underlying the findings in their manuscript fully available?

Reviewer #1: Yes

Reviewer #2: Yes

Reviewer #3: Yes

4. Is the manuscript presented in an intelligible fashion and written in standard English?

Reviewer #1: Yes

Reviewer #2: Yes

Reviewer #3: Yes

5. Review Comments to the Author

Reviewer #1: The authors present A LOT of data to show that subterranean clover is morphologically highly diverse. This claim is supported by the results.

General comments: the terms 'diversity' and 'variation' seem to be used interchangeably, please be more clear, also in the hypotheses. Presentation of the results of the statistical analyses is not always clear. The manuscript is fairly easy to read, even though there are lots of technical details, in my opinion the English is good.

Line 193 and further on: did you account for an edge effect?

Line 238: Explain how are the parameter values determined?

Figures 2-6: some kind of error measure would greatly improve these figures. Figs 3-6 are stacked bar charts, whereas fig2 is a side-by-side bar chart, consistency could be improved. Fig 5 and 6 lack a legend.

Line 348: in addition to the reference please state how you calculated heritability.

Line 355 and table 3: presentation of this result not very clear, all differences ere significant, differences between what? Same for line 365 and further on. flowering time is different among ALL 125 genotypes? Would be more clear if F and p-values would be mentioned.

Figure 2, legend: 'values are the mean of four replicate single plants' If measurements are done on four out of 97 or 28 plants This should be specified in the methods sections.

Table 5: maybe color-code significance for easier reading/interpretation of the table?

Reviewer #2: The authors present a comprehensive study on a core collection of subterranean clover species consisting of 97 lines, which accounted for approximately 80% of the genetic diversity of the global subterranean clover collection along with 28 Australian cultivars. Assessments were made based on morphological marker traits, agro-morphological traits and phytoestrogen levels in subterranean sp. Trait associations were evaluated based on correlation. It is this reviewer’s opinion that the manuscript presents a comprehensive body of work that will contribute towards developing new cultivars better-suited for establishment in southern Australian agro-climatic conditions. The manuscript is organised in a logical manner, in sound English with only minor typographical errors.

General comment:

1. Please note that throughout the manuscript the abbreviation ssp is misrepresented as spp, intermittently and requires correction.

Specific comments:

1. Line 150: should read “DNA was extracted from each member of this subset and an 20 additional Australian cultivars. Forty eight SSR primers spread across each of the eight subterranean clover chromosomes….”

2. Line 306: It is best to be more descriptive with the method used. For example the “alcohol” used in this method is ethanol.

3. While the method for quantifying isoflavones used in this study is valid, the authors should further identify the limitations in using thin layer chromatographic techniques due to the semi quantitative nature and elude to the fact that more precise LC-MS approaches can be useful for absolute quantification of isoflavones. It is logical that the choice of method was due to the high-throughput nature and lower cost associated with TLC, nevertheless I suggest the authors acknowledge that newer technologies can be utilised.

4. The authors observed associations between several morphological markers and environmental factors and suggest that the adaptions are possibly associated with the regulation of genes or interaction between several genes. The authors should also however elude to possibility of epigenetic regulation leading to observed traits.

Reviewer #3: Dear author, the manuscript is very interesting and technically sound.

I think it is an important study of morphological diversity of Trifolium subterraneum compiling a large data set of several countries. It was possible to link morphological marker traits with environmental factors, being a great contribute to this research subject.

However, I suggest an improvement of the graphic quality of figures 4 and 5 (legends…) and a revision of typos: see, e.g., line 109 “Chile, Iran Portugal and South” should be replaced by ““Chile, Iran, Portugal and South”.

Kind regards.

6. PLOS authors have the option to publish the peer review history of their article (what does this mean?). If published, this will include your full peer review and any attached files.

Reviewer #1: No

Reviewer #2: No

Reviewer #3: No

---

## [Author Response · Author response to Decision Letter 0]

2 Dec 2019

In this response, Line numbers refer to those in the original submitted manuscript.

Review 1

General comments 

The authors present A LOT of data to show that subterranean clover is morphologically highly diverse. This claim is supported by the results. The manuscript is fairly easy to read, even though there are lots of technical details. In my opinion the English is good.

The terms 'diversity' and 'variation' seem to be used interchangeably, please be more clear, also in the hypotheses. 

The term ‘variation’ has been replaced with ‘diversity’ throughout the manuscript for consistency.

Presentation of the results of the statistical analyses is not always clear. 

Concerns with the presentation of some individual statistical analyses raised by Reviewer 1 have been addressed in the comments below. The other reviewers did not raise any concerns about the statistical analyses.

Specific comments attached to manuscript

Line 193 and further on: did you account for an edge effect?

The issue of potential edge effects has been clarified with the additional sentence “This spacing was sufficient to allow plants to grow without competition for light and moisture and prevented edge effects”. It is also clarified in the following sentence referring to additional plants to calibrate shoot dry weight measurements by adding “using the same spacing”.

Line 238: Explain how are the parameter values determined?

Determination of plant dry weight has been explained better by referring specifically to the calibration plants used to estimate the regression equation.

Figures 2-6: some kind of error measure would greatly improve these figures. 

Figures 2-6 are frequency distributions of qualitative, rather than quantitative, traits and consequently, there are no appropriate statistical error measurements that can be displayed. Titles for the vertical axes of each of these graphs have been updated for greater clarity to indicate that data in each case represents numbers of genotypes in each category.

Figs 3-6 are stacked bar charts, whereas fig2 is a side-by-side bar chart, consistency could be improved. 

Figure 2 has been changed to a stacked bar chart for consistency.

Fig 5 and 6 lack a legend.

Legends for figures 5 and 6 have been removed from the caption and placed directly onto the figures for greater clarity.

Line 348: in addition to the reference please state how you calculated heritability.

The calculation of heritability has been clarified by the following sentence: “Broad-sense heritability (H2) was estimated for each quantitative trait as the percentage of the phenotypic variance attributable to genotypic variance, in accordance with Falconer et al. [38]”.

Line 355 and table 3: presentation of this result not very clear, all differences were significant, differences between what? Same for line 365 and further on. 

Differences were highly significant for all traits measured among the 125 genotypes at a level of P ≤ 0.001. This is explained in the caption for Table 3. The footnote “ *P ≤ 0.05, **P ≤ 0.01, ***P ≤ 0.001” at the bottom of Table 3 is superfluous and a potential source of confusion for this table and has been removed.

Flowering time is different among ALL 125 genotypes? Would be more clear if F and p-values would be mentioned.

A value of P ≤ 0.001 for flowering time is given in Table 3, indicating a significant difference between the 125 genotypes for flowering time. The P value (P ≤ 0.001) has been added to the text in Line 365, as suggested, for greater emphasis.

Figure 2, legend: 'values are the mean of four replicate single plants' If measurements are done on four out of 97 or 28 plants This should be specified in the methods sections.

The methodology of measuring flowering time of each genotype on four single plants is explained in detail in the Materials and Methods. The sentence “Values are the mean of four replicate single plants” has been removed from the caption for Figure 2 for simplicity. Information has also been removed from subsequent captions, where it is explained in detail in the Materials and Methods and is not required for interpretation of the figures.

Table 5: maybe color-code significance for easier reading/interpretation of the table?

The different levels of significance have been colour-coded for each trait correlation in Table 5, as suggested by the reviewer. The same colour-coding has also been applied to Tables 6-9. The colour scheme can be modified (or deleted) to suit the journal, according to the editor’s suggestions.

Review 2

General comments 

The authors present a comprehensive study on a core collection of subterranean clover species consisting of 97 lines, which accounted for approximately 80% of the genetic diversity of the global subterranean clover collection along with 28 Australian cultivars. Assessments were made based on morphological marker traits, agro-morphological traits and phytoestrogen levels in subterranean sp. Trait associations were evaluated based on correlation. It is this reviewer’s opinion that the manuscript presents a comprehensive body of work that will contribute towards developing new cultivars better-suited for establishment in southern Australian agro-climatic conditions. The manuscript is organised in a logical manner, in sound English with only minor typographical errors.

Throughout the manuscript the abbreviation “ssp.” is misrepresented as “spp”, intermittently and requires correction.

The abbreviation for subspecies has been corrected to “ssp.” throughout the manuscript.

Specific comments attached to manuscript

Line 150: should read “DNA was extracted from each member of this subset and an 20 additional Australian cultivars. Forty eight SSR primers spread across each of the eight subterranean clover chromosomes….”

These two sentences have been corrected as advised.

Line 306: It is best to be more descriptive with the method used. For example the “alcohol” used in this method is ethanol.

The term “alcohol” has been corrected to “ethanol”.

While the method for quantifying isoflavones used in this study is valid, the authors should further identify the limitations in using thin layer chromatographic techniques due to the semi quantitative nature and elude to the fact that more precise LC-MS approaches can be useful for absolute quantification of isoflavones. It is logical that the choice of method was due to the high-throughput nature and lower cost associated with TLC, nevertheless I suggest the authors acknowledge that newer technologies can be utilised.

This point has been addressed by adding the following sentences: “This method was used due to its high-throughput and low cost, while recognising its semi-quantitative nature imposes limitations in precision. This method has also been previously used in similar genetic diversity studies by Nichols et al. [16] and Ghamkhar et al. [31]”.

The authors observed associations between several morphological markers and environmental factors and suggest that the adaptions are possibly associated with the regulation of genes or interaction between several genes. The authors should also however elude to possibility of epigenetic regulation leading to observed traits.

We have incorporated the reviewer’s comment on the possible role of epigenetic inheritance by adding the sentence to the Discussion “Epigenetic inheritance may also be important and needs further investigation”. The reference “Quadrana L, Colot V. Plant Transgenerational Epigenetics. Ann. Rev. Genet. 2016; 50: 467-491” has also been added.

Review 3

General comments 

The manuscript is very interesting and technically sound. I think it is an important study of morphological diversity of Trifolium subterraneum compiling a large data set of several countries. It was possible to link morphological marker traits with environmental factors, being a great contribute to this research subject.

Specific comments attached to manuscript

I suggest an improvement of the graphic quality of figures 4 and 5 (legends…) 

The graphic quality of Figures 3, 4 and 5 have been improved. Legends for Figures 5 and 6 have been placed directly onto the figures for greater clarity. Titles for the vertical axes of each graph in Figures 2-6 have been updated to indicate that data in each case represents numbers of genotypes in each category.

I suggest a revision of typos 

The manuscript has been proofread again and typos have been corrected.

Line 109 “Chile, Iran Portugal and South” should be replaced by ““Chile, Iran, Portugal and South”.

This sentence has been corrected.

---

## [Editor Report · Decision Letter 1]

6 Dec 2019

PONE-D-19-26944R1

Morphological diversity within a core collection of subterranean clover (Trifolium subterraneum L.): lessons in pasture adaptation from the wild

PLOS ONE

Dear Dr Nichols,

Thank you for submitting your manuscript to PLOS ONE. After careful consideration, we feel that it has merit but does not fully meet PLOS ONE’s publication criteria as it currently stands. Therefore, we invite you to submit a revised version of the manuscript that addresses the points raised during the review process.

-I observe that there are 9 tables and 7 figures on the main text. I guess they are too many for a traditional research paper. Please consider to replace some figs or tables in supplementary material.

-Improve the quality of the figures: add major and minor marks on the axes, remove or reduce the empty spaces (ex: the empty spaces between vertical bars). 

We would appreciate receiving your revised manuscript by Jan 20 2020 11:59PM. To enhance the reproducibility of your results, we recommend that if applicable you deposit your laboratory protocols in protocols.io, where a protocol can be assigned its own identifier (DOI) such that it can be cited independently in the future. For instructions see: http://journals.plos.org/plosone/s/submission-guidelines#loc-laboratory-protocols

We look forward to receiving your revised manuscript.

Kind regards,

Sergio Rossi

Academic Editor

PLOS ONE

---

## [Author Response · Author response to Decision Letter 1]

23 Dec 2019

Response to Academic editor’s comments

1. I observe that there are 9 tables and 7 figures in the main text. This is too many for a traditional research paper. Please consider replacing some figures or tables in the Supplementary material.

I have reduced the number of figures (four) and tables (two) from the main text and moved them to the Supplementary material, as suggested. This now means the main text has three figures and seven tables. Please advise if this is still too many. In particular, the original Table 1 is now S2 Table, original Table 2 is now S3 Table, original Fig 3 is now S2 Fig, original Fig 4 is now S3 Fig, original Fig 5 is now S4 Fig and original Fig 6 is now S5 Fig. The text has been modified to take account of these changes.

2. Improve the quality of the figures: add major and minor marks on the axes, remove or reduce the empty spaces (ex: the empty spaces between vertical bars).

The graphs have been improved, as suggested. Major and minor tick marks have been added to the axes and the spaces between vertical bars in the bar graphs have been markedly reduced.

---

## [Editor Report · Decision Letter 2]

27 Dec 2019

Morphological diversity within a core collection of subterranean clover (Trifolium subterraneum L.): lessons in pasture adaptation from the wild

PONE-D-19-26944R2

Dear Dr. Nichols,

We are pleased to inform you that your manuscript has been judged scientifically suitable for publication and will be formally accepted for publication once it complies with all outstanding technical requirements.

With kind regards,

Sergio Rossi

Academic Editor

PLOS ONE
---

## [Editor Report · Acceptance letter]

31 Dec 2019

PONE-D-19-26944R2 

Morphological diversity within a core collection of subterranean clover (Trifolium subterraneum L.): lessons in pasture adaptation from the wild 

Dear Dr. Nichols:

I am pleased to inform you that your manuscript has been deemed suitable for publication in PLOS ONE. Congratulations! Your manuscript is now with our production department. 

With kind regards,

on behalf of

Prof. Sergio Rossi 

Academic Editor

PLOS ONE